# GraphPulse: Topological Representations for Temporal Graph Property Prediction

**Kiarash Shamsi[1]**     **Farimah Poursafaei[2,3]**     **Shenyang Huang [2,3]**     **Tran Gia Bao Ngo[1]**
**Baris Coskunuzer[4]**     **Cuneyt Gurcan Akcora[5]**
[1]Department of computer science, University of Manitoba, [2]Mila - Quebec AI Institute,
[3]School of Computer Science, McGill University, [4]University of Texas at Dallas,
[5]AI Institute - University of Central Florida
[shamsik1@myumanitoba.ca, farimah.poursafaei@mila.quebec,
shenyang.huang@mail.mcgill.ca, ngot1@myumanitoba.ca,
coskunuz@utdallas.edu, cuneyt.akcora@ucf.edu]

## Abstract

Many real-world networks evolve over time, and predicting the evolution of such networks remains a challenging task. Graph Neural Networks (GNNs) have shown empirical success for learning on static graphs, but they lack the ability to effectively learn from nodes and edges with different timestamps. Consequently, the prediction of future properties in temporal graphs remains a relatively under-explored area. In this paper, we aim to bridge this gap by introducing a principled framework, named GraphPulse. The framework combines two important techniques for the analysis of temporal graphs within a Newtonian framework. First, we employ the Mapper method, a key tool in topological data analysis, to extract essential clustering information from graph nodes. Next, we harness the sequential modeling capabilities of Recurrent Neural Networks (RNNs) for temporal reasoning regarding the graph's evolution. Through extensive experimentation, we demonstrate that our model enhances the ROC-AUC metric by 10.2% in comparison to the top-performing state-of-the-art method across various temporal networks. We provide the implementation of GraphPulse at `https://github.com/kiarashamsi/GraphPulse`.

## 1 Introduction

Real-world interaction networks, such as financial and cryptocurrency networks, experience continuous evolution due to the emergence of new transactions and users. Predicting the dynamic changes in the graph structure of these networks over time poses a significant challenge. While Graph Neural Networks have proven effective for learning graph representations in static graphs (Xu et al., 2018), temporal graphs differ significantly because nodes and edges continuously change at various timestamps. Classical GNNs are not well-suited for temporal graphs as they do not effectively utilize crucial temporal information, such as alterations in the graph structure over time. Hence, there is a pressing need for a novel approach to adapt GNNs successfully to temporal graph settings, giving rise to a new field known as Temporal Graph Neural Networks (TGNNs).

In order to understand the evolution of a temporal graph, it is essential to capture the global graph structure across time points and then quantify the changes and evolution that the graph has undergone. For instance, in a subsequent time step, the graph might introduce a new node or establish an edge between two existing nodes to significantly shorten the graph diameter. While random graph models such as the Erdos-Renyi graph (Erdős & Rényi, 1960) and the Stochastic Block Models (SBMs) (Holland et al., 1983) have been studied extensively in the static graph literature (Barabási, 2013), there is few theoretical work studying the evolution of temporal graphs in the transition space of graph generative models.

We aim to close this gap and introduce *GraphPulse* to efficiently capture the evolving structure of a graph over time in an innovative temporal framework. Our principled approach posits two hypotheses. First, we suggest that the evolution of a graph can be represented as a temporal trajectory

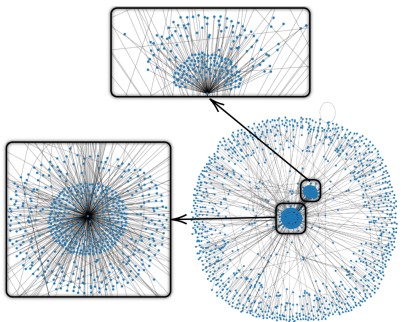 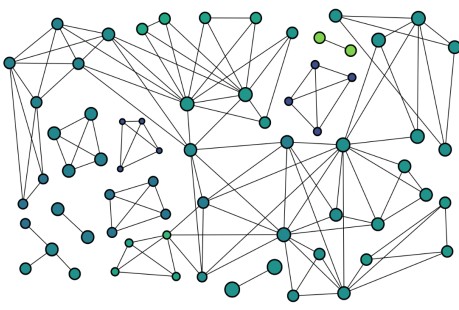

(a) Daily transaction graph of an Ethereum token    (b) TDA Mapper network of the graph in 1a.

Figure 1: **Illustration of TDA Mapper Network.** The daily transaction graph of an Ethereum token (a) is transformed into a concise Mapper network (b), where nodes represent clusters of token investors, and edges are assigned weights based on the shared number of nodes between clusters. Node sizes indicate cluster sizes and node colors are Disconnected components, highlight node groups with divergent graph characteristics compared to the rest of the nodes.

in a Newtonian phase space (Newton, 1833). Second, we propose that a topological technique, Mapper (Singh et al., 2007), can be effectively employed to model this trajectory. Mapper generates concise and principled visual representations of data, as exemplified in Figure 1. As a third step, we utilize the topological information in a Recurrent Neural Network to predict a graph feature in the future.

We demonstrate the effectiveness of our approach in two types of experiments. First, we empirically prove that Mapper networks can capture temporal trajectories effectively by considering two common phase spaces for graphs: Erdos-Renyi graphs (Erdős & Rényi, 1960) and Stochastic Block Models (SBMs) (Holland et al., 1983; Barabási, 2013). Subsequently, we introduce a unique task focused on predicting graph properties in dynamic graphs and demonstrate that the GraphPulse outperforms existing Graph Neural Network and Temporal Graph Neural Network models in social and cryptocurrency networks. In nine temporal network scenarios within these contexts, our model consistently surpasses state-of-the-art approaches in eight instances.

**Our Contributions:**

- We present a novel framework, GraphPulse, for analyzing the evolution of temporal graphs, grounded in phase space transition. We empirically show that the TDA Mapper algorithm is highly effective at capturing the evolution of temporal graphs in the phase space.

- We introduce a unique and valuable task involving the prediction of temporal graph properties in the future. GraphPulse takes advantage of sequential modeling to capture temporal variations, and it effectively integrates node feature information into the model using the TDA Mapper algorithm.

- We create seven original cryptoasset networks for the temporal graph property task and publish them to serve as temporal benchmark datasets for future research.

- Empirically, GraphPulse significantly outperforms state-of-the-art temporal graph learning models in eight out of nine networks from both social and transaction network domains.

## 2 BACKGROUND AND RELATED WORK

In this section, we first define a temporal or dynamic graph and its property. Then, we give the related work for Topological Data Analysis on graphs and temporal graph neural networks.

**Discrete Time Dynamic Graphs.** A Discrete Time Dynamic Graph (DTDG) is defined as a series of snapshot graphs denoted as $\mathcal{G} = \{\mathcal{G}_{t_1}, \mathcal{G}_{t_2}, \ldots, \mathcal{G}_{t_n}\}$, each $\mathcal{G}_{t_i} = (\mathcal{V}_i, \mathcal{E}_i)$ represents the graph at time step $i$ and $\mathcal{V}_i$ and $\mathcal{E}_i \subseteq \mathcal{V}_i \times \mathcal{V}_i$ are the set of nodes and edges at time $i$ respectively. The graph can be directed or undirected and the Nodes can also have attributes.

**Temporal Graph Property Prediction.** The goal of the Temporal Graph Property Prediction task is to predict a specific property of a temporal graph over a future time interval in a Discrete Time

Dynamic Graph. Formally, given a DTDG $\mathcal{G}$, we define a target time interval $[n + \delta_1, n + \delta_2]$, where $\delta_1$ and $\delta_2$ are non-negative integers with $\delta_1 \leq \delta_2$. The objective is to predict the values of the chosen graph property within the specified future interval $[n + \delta_1, n + \delta_2]$. Examples of graph properties include metrics such as graph density, the count of newly added edges, and the cumulative weight of edges across the graph. Other temporal graph properties include temporal global efficiency, temporal-correlation coefficient, and temporal betweenness centrality can also be explored in the future (Nicosia et al., 2013).

## 2.1 TDA & MAPPER

Existing methods in Topological Data Analysis can be categorized into two main approaches: persistent homology (PH) and Mapper. PH studies the evolution of topological features (connected components, loops, voids, etc.) of a dataset at various spatial resolutions Kyriakis et al. (2021). PH has recently been utilized as a powerful feature extractor and combined with deep learning methods (Hofer et al., 2020; Carrière et al., 2020; Chen et al., 2021; Horn et al., 2021) in node and graph classification tasks. However, PH-based methods face limitations in their application to large networks due to their high computational complexity.

The *"Mapper"* method was introduced by Singh et al. (2007) and further elaborated by Carlsson (2009). This approach serves the purpose of converting complex, high-dimensional data into a more comprehensible and coordinate-independent graphical representation to adeptly capture the underlying topological attributes of the data (Van Veen et al., 2019; Tauzin et al., 2021). Over the past decade, the Mapper technique has found fruitful applications in various domains (Lum et al., 2013; Kamruzzaman et al., 2019; Zhou et al., 2021) yielding remarkable outcomes. More recently, the Mapper algorithm found utility in the domain of graph representation learning, as highlighted in the works of Hajij et al. (2018); Bodnar et al. (2021). However, to the best of our knowledge, this is the first work to utilize Mapper in temporal Graph ML.

## 2.2 TEMPORAL GRAPH NEURAL NETWORKS

Temporal Graph Neural Networks (TGNNs) have shown promising performance on tasks such as link prediction and node classification (Rossi et al., 2020; Souza et al., 2022; Pareja et al., 2020; You et al., 2022; Huang et al., 2023c). Kazemi et al. (2020) categorize temporal graphs into Discrete Time Dynamic Graphs (DTDGs) (You et al., 2022; Gao & Ribeiro, 2022; Pareja et al., 2020; Hajiramezanali et al., 2019; Yang et al., 2021) and Continous Time Dynamic Graphs (Xu et al., 2020; Rossi et al., 2020; Poursafaei et al., 2022; Souza et al., 2022; Luo & Li, 2022; Jin et al., 2022). In this work, we focus on DTDGs where the temporal graph is modelled as a sequence of snapshots. Although tasks like link prediction and anomaly detection in temporal graphs have gained popularity, there is still a lack of exploration in graph-level tasks specifically tailored for temporal graphs. One related task is network change point detection that aims to detect time points where the temporal graph undergoes drastic structural changes (Huang et al., 2020; Miller & Mokryn, 2020; Wang et al., 2017; Huang et al., 2023b;a). While change point detection focus on detecting anomaly from observed network structure, we focus on predicting future graph properties such as network growth or shrinkage in this work.

The study of the expressiveness of GNNs often employs the Weisfeiler-Lehman (WL) test which tests for graph isomorphism (Xu et al., 2018; Maron et al., 2019; Cotta et al., 2021). On temporal graphs, recent work (Gao & Ribeiro, 2022; Souza et al., 2022) attempts to extend the WL test to temporal graphs as converting them into static representations. However, real-world temporal networks typically deviates from the graph isomorphism based analysis due to the continuous addition of nodes and edges at different timestamps. We argue that an alternative framework is necessary to analyze the practical effectiveness of GNNs. We introduce a novel concept, called *phase space*, for temporal graphs to address this need in Section 4.

## 3 MAPPER AND TOPOLOGICAL GRAPH REPRESENTATION

In this section, we explain the first component of GraphPulse, the Mapper, and its application to a single graph. The core principle underlying Topological Data Analysis revolves around uncovering latent data patterns through systematic analysis of data shapes, which are measured across various

resolution scales (Chazal & Michel, 2021; Dey & Wang, 2022). TDA brings forth a range of compelling advantages, notably pertinent within the realm of graph machine learning. Primarily, TDA scrutinizes data shapes in a manner free from coordinate constraints, enabling systematic comparison of patterns derived from diverse data collection frameworks. This adaptability proves invaluable, accommodating temporal scenarios such as comparing graphs in distinct time periods.

In this work, we employ the Mapper method as introduced by (Singh et al., 2007; Carlsson, 2009). For a compact topological space $\mathcal{X}$ and a real-valued function $f : \mathcal{X} \to \mathbb{R}$, the Mapper algorithm provides a general framework to study the topological changes in $\mathcal{X}$ with respect to the function $f$, which is commonly referred to as a *filter function* or *lens*. The choice of lens is crucial in Mapper construction as various lenses provide distinct insights on the data (Singh et al., 2007; Dey & Wang, 2022). In practice, $\mathcal{X}$ is mostly a point cloud in $\mathbb{R}^N$, and $f$ is a function from $\mathbb{R}^N$ to $\mathbb{R}$ representing some domain information of the data. The output of the Mapper algorithm is the Mapper network, which is considered a meaningful summary of the data, representing clusters and relations between the clusters in the data.

To define the Mapper network, we need to explain the nerve of a cover. Let $\mathcal{X}$ be a point cloud in $\mathbb{R}^N$. A *cover* of $\mathcal{X}$ is a set of open sets in $\mathbb{R}^N$, $\mathcal{U} = \{U_i\}_{i=1}^m$ with $\mathcal{X} \subset \bigcup_i U_i$. The 1D nerve of $\mathcal{U}$ is a graph and is denoted as $\eta_1(\mathcal{U})$. Each node $v_i$ in $\eta_1(\mathcal{U})$ represents a cover element $U_i$, and an edge exists between two nodes $v_i$ and $v_j$ if $U_i \cap U_j$ is nonempty for the corresponding cover elements. The appendix Figure 4-a gives an example in which $\mathcal{X}$ is a 2D point cloud and the cover $\mathcal{U}$ of $\mathcal{X}$ consists of a collection of rectangles on the plane.

While it is possible to use multiple scalar functions, to keep the exposition clear, we describe the Mapper construction with a single scalar function $f : \mathcal{X} \to \mathbb{R}$. We start with a finite cover of $f(\mathcal{X}) \subset \mathbb{R}$ using intervals, that is, a cover $\mathcal{I} = \{I_k\}_{k=1}^n$ of $f(\mathcal{X}) \subset \mathbb{R}$ such that $f(\mathcal{X}) \subset \bigcup_k I_k$, see Figure 4. This induces a cover $\mathcal{U}$ of $\mathcal{X}$ by considering the clusters induced by points in $f^{-1}(I_k)$ for each $k$ as a cover element. The 1D *Mapper network* of $(\mathcal{X}, f)$, denoted as $\Gamma$, is nothing but the 1D nerve of $\mathcal{U}$, i.e. $\Gamma_\mathcal{X} := \eta_1(\mathcal{U})$.

For a point cloud $\mathcal{X}$, several choices are to be made to compute the Mapper network $\Gamma_\mathcal{X}$. The first and most important one is the lens function $f : \mathcal{X} \to \mathbb{R}$. Another choice is the clustering method for the point cloud. Finally, there are two tuning parameters. The first one is called *resolution*, which is the number of intervals $\{I_k\}$ to cover $f(\mathcal{X})$. The second one is called *gain* which is the percentage of overlaps of these intervals. Note that increasing the resolution gives a finer summary by increasing the number of nodes in the Mapper network and makes the clusters smaller. On the other hand, increasing gain adds more relation (edges) between the nodes (clusters). In Figure 4, the resolution is 6, and the gain is the fixed intersection amount (e.g., 20%) between the intervals $I_k$ and $I_{k+1}$.

Note that the Mapper construction can be generalized to higher dimensions by choosing the lens function $f$ as a multivariate function, i.e., $f : \mathcal{X} \to \mathbb{R}^d$ (for $d \geq 2$). In most cases, $d = 2$, and the resulting Mapper network is referred to as a *2D Mapper network*, where the corresponding cover elements of $\mathbb{R}^2$ become rectangles.

**Mapper for graphs.** For any graph $\mathcal{G}$, we take the set of nodes $\mathcal{V} = \{v_1, \ldots, v_n\}$ and use their feature vectors $\mathbf{X}_i \in \mathbb{R}^N$ as our point cloud $\mathcal{X}_\mathcal{G} = \{\mathbf{X}_1, \ldots, \mathbf{X}_n\} \subset \mathbb{R}^N$. Then, we obtain and use the Mapper network $\Gamma_\mathcal{G}$ of this point cloud $\mathcal{X}_\mathcal{G}$ as a summary of the feature information stored in the graph $\mathcal{G}$ in our approach. The hyperparameters for our Mapper construction (lens function, clustering method, resolution, and gain) are discussed in Section 6.

## 4    TRAJECTORIES OF TEMPORAL GRAPHS

In this section we explain the second component of the GraphPulse, where we represent graphs in a space, enabling us to directly compare their Mapper representations. The concepts of phase space and trajectory in this section will serve as the foundation on which we will build a new temporal framework.

We propose a topological model for describing the progression of temporal graphs. Specifically, we posit the presence of a phase space about a family of temporal graphs, wherein the sequence of temporal graphs gives rise to a trajectory (path) within this phase space. Subsequently, for each graph,

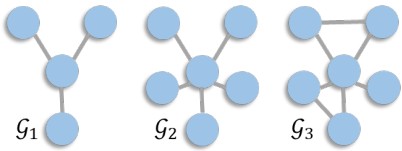 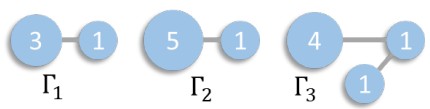

(a) The graph $\mathcal{G}_1$ has four nodes and three edges. In $\mathcal{G}_2$, the graph gains two more nodes and two edges. However, only in $\mathcal{G}_3$ do two triangular edges emerge in the graph.

(b) Induced TDA Mapper networks $\Gamma_1, \Gamma_2$ and $\Gamma_3$ for $\mathcal{G}_1$, $\mathcal{G}_2$ and $\mathcal{G}_3$ by using degree as the node feature. Mapper node sizes (denoted by numbers) indicate cluster sizes.

Figure 2: **Evolution of Graph Complexity.** Mapper representations (b) of snapshot graphs (a) where the Mapper lens uses the number of neighbors.

we induce its topological summary through the Mapper algorithm, which enables us to monitor the evolution of these Mapper networks over time.

## 4.1 TRAJECTORY IN A PHASE SPACE

The core idea for our temporal framework is inspired by Newton's laws of motion (Newton, 1833) that specifies how a system's variables change over time in response to forces acting upon it. The idea of *phase space* emerges from Newtonian mechanics as a natural extension of these principles (Carroll, 2022). A phase space $\mathcal{P}$ refers to a multi-dimensional space where the state of a system is represented by a set of coordinates corresponding to its variables. In our context, one can imagine this notion as a correspondence map $\varphi : \mathcal{P} \rightarrow \mathfrak{S}$ where $\mathfrak{S}$ represents the space of graphs, where any point $x$ in the phase space $\mathcal{P}$ represents a graph $\varphi(x) = \mathcal{G}_x \in \mathfrak{S}$. Hence, **each point in the phase space represents a graph instance**: a specific configuration of the system's variables, capturing its instantaneous or momentary state (see Appendix Figure 8 for an example). In a dynamical system, the trajectories of a system's evolution over time can be visualized as paths in the phase space. This concept is crucial for analyzing the behavior, stability, and evolution of complex systems by observing how their states change in response to various influences or forces.

Formally, let $\mathfrak{S}_\mathcal{P}$ denote a family of graphs parameterized by the parameter space $\mathcal{P}$. To simplify our notation, we will employ Erdős-Rényi graphs as an illustrative example, where $\mathcal{P} = \mathbb{N} \times [0, 1]$ (see Appendix Figure 6). In this context, the generative parameters are $(n, p) \in \mathcal{P}$, where $n$ signifies the number of nodes, and $p$ signifies the probability of an edge. This notation can be straightforwardly extended to encompass scenarios involving more than two parameters. Consequently, for $x \in \mathcal{P}$, $\mathcal{G}_x \in \mathfrak{S}_\mathcal{P}$ corresponds to the graph within $\mathfrak{S}_\mathcal{P}$ associated with that parameter $x$. We assume a one-to-one correspondence between the elements of $\mathcal{P}$ and the graphs within $\mathfrak{S}_\mathcal{P}$. Consider a dynamic graph family $\{\widehat{\mathcal{G}}^i\}$ whose instances $\mathcal{G}^i_{t_j}$ are in $\mathfrak{S}_\mathcal{P}$, i.e., $\widehat{\mathcal{G}}^i = (\mathcal{G}^i_{t_1}, \mathcal{G}^i_{t_2}, \ldots, \mathcal{G}^i_{t_n})$ an ordered sequence of graphs. By using one-to-one correspondence, we induce a set of trajectories $\{\widehat{\mathbf{x}}^i\}$ with $\widehat{\mathbf{x}}^i = (\mathbf{x}^i_1, \mathbf{x}^i_2, \ldots, \mathbf{x}^i_n)$, i.e., $\mathbf{x}^i_j \in \mathcal{P}$ is the the $j^{th}$ step of the $i^{th}$ trajectory which corresponds to the graph $\mathcal{G}^i_{t_j} \in \mathfrak{S}$.

For any point $\mathbf{x}^i_j \in \mathcal{P}$, we define a vector $\nu^i_j = \mathbf{x}^i_{j+1} - \mathbf{x}^i_j$ where $1 \le j < n$. The pair $(\mathbf{x}^i_j, \nu^i_j)$ defines a *discrete dynamical system* in the parameter space $\mathcal{P}$ (Galor, 2007). Our idea is to learn this induced discrete dynamical system in the parameter space and obtain some useful feature vectors for future instances. In other words, by using the prediction of future trajectory points of $\mathbf{x}^i$ in parameter spaces, we extract useful feature vectors to predict the future instances of dynamic network $\widehat{\mathcal{G}}^i$.

## 4.2 TRAJECTORY IN THE TOPOLOGICAL SPACE

Consider a temporal trajectory $\widehat{\mathbf{x}} = (\mathbf{x}_0, \ldots, \mathbf{x}_n)$ with $\mathbf{x}_i \in \mathcal{P}$, i.e., $\widehat{\mathbf{x}}$ is an $n$-step trajectory with the initial point $\mathbf{x}_0$ to the final point $\mathbf{x}_n$ passing through intermediate points $\mathbf{x}_1, \mathbf{x}_2, ..., \mathbf{x}_{n-1}$. While we move on the phase space $\mathcal{P}$ through $\widehat{\mathbf{x}}$, the original graph family defines a corresponding trajectory $\widehat{\mathcal{G}} = (\mathcal{G}_0, \ldots, \mathcal{G}_n)$ in the space of graphs $\mathfrak{S}_\mathcal{P}$. Similarly, for the same trajectory $\widehat{\mathbf{x}} \subset \mathcal{P}$, we induce the corresponding TDA Mapper trajectory $\widehat{\Gamma}$ in the space of graphs where each $\Gamma_i$ is the TDA Mapper network of $\mathcal{G}_i$ by using the induced features (Section 3). We give an illustration of our TDA Mapper networks on a toy example using only one node feature (node degree) in Figure 2.

We hypothesize that trajectories can be captured and analyzed efficiently by using Mapper networks of the snapshot graphs. To test our hypothesis, we run extensive experiments in two well-known phase space settings, i.e., Erdős-Rényi and Barabási-Albert settings in Appendix A. For a given trajectory $\widehat{\mathbf{x}} = (\mathbf{x}_0, \dots, \mathbf{x}_n)$ in the phase space $\mathcal{P}$, we obtain the corresponding original graph trajectory $\{\mathcal{G}_i\}$ and the corresponding TDA Mapper trajectory $\{\Gamma_i\}$. We compare the structural changes in the original graph trajectory and changes in the TDA Mapper trajectory. In order to measure the structural similarity of graphs, we use a similarity measure induced from graph Laplacian eigenvalues (Koutra et al., 2011). Specifically, our results in Appendix Figure 9 show that Mapper based trajectory encodes neighborhood in the phase space more efficiently. Furthermore, as we show in Section 6, our experimental findings demonstrate a significant enhancement in the predictive capacity of our model within real-world networks due to these topological summaries.

### 4.3 Advantages of TDA Mapper networks

Our approach introduces a novel perspective to representing and visualizing the evolution of dynamic systems in parameter space. By doing so, we can gain several advantages over using traditional snapshot graphs for trajectory analysis, as follows.

**Topological Features.** Mapper networks focus on essential topological features, providing a nuanced view of graph evolution under changing parameters. Consider Figure 2a that shows three graphs of increasing node counts and connectivity. As we transition from $\mathcal{G}_1$ to $\mathcal{G}_2$, the sole modification involves introducing two nodes while retaining the pattern of satellite nodes linking to a crucial central node. Such a scenario is common on blockchain networks where addresses of blockchain exchanges distribute tokens in airdrops (Victor, 2020). The Mapper encapsulates that $\mathcal{G}_1$ and $\mathcal{G}_2$ are similar by creating a network of size two with a connecting edge for both $\Gamma_1$ and $\Gamma_2$. Retaining such essential aspects in networks becomes particularly significant when addressing complex interactions that might not be evident through simple node attributes (e.g., degree) alone.

**Compressed Representation.** Mapper networks compress complex data into compact yet meaningful representations, enhancing visualization and robustness against data fluctuations. In Figure 2a, we can add another triangle $\triangle$ to $\mathcal{G}_3$ centered on the existing center graph node, but Mapper would just include the two new graph nodes to the bottom Mapper node (i.e., cluster) of $\Gamma_3$ without adding a new mapper node.

**Multi-Resolution.** Mapper's multi-resolution capability enables a detailed exploration of trajectories, revealing fine details and broader trends. As the trajectory moves through parameter space, we can identify clusters (i.e., network nodes) and connections, unveiling significant regions and transitions often hidden in raw snapshot graphs. With clear node and edge features, the resulting visualizations offer valuable insights into dynamic system behavior across parameter trajectories.

Due to space limitations, we report a comparative analysis of snapshot graphs and their Mapper networks in the Appendix. In particular, we demonstrate that Mapper networks exhibit greater stability than the original snapshot graphs when subjected to minor perturbations within the phase space. This observation underscores the considerable utility of Mapper summaries in predicting trajectories. We further validate our hypothesis regarding Mapper efficiency through an empirical evaluation of a novel task of predicting graph properties as we discuss in Section 6.

## 5 GraphPulse

So far, we have explained how the Mapper-based topological representations can help us track graph trajectories in the phase space. In this section, we describe the overall flow of GraphPulse and detail the third step: sequential modeling.

Our methodology comprises a sequence of three distinct phases, illustrated in the visual representation provided in Figure 3: partitioning, topological learning, and sequential modeling. The partitioning step involves the conversion of input data into snapshot graphs denoted as $\widehat{\mathcal{G}} = \{\mathcal{G}_{t_1}, \mathcal{G}_{t_2}, \dots, \mathcal{G}_{t_n}\}$. In this work, we adopt 24-hour snapshots, although our model remains applicable to snapshots of varying durations.

In the topological learning step, we construct TDA Mapper representations for graphs $\mathcal{CT}_1, \dots, \mathcal{CT}_n$, each corresponding to a graph within $\{\widehat{\mathcal{G}}\}$. To achieve this, we first extract graph

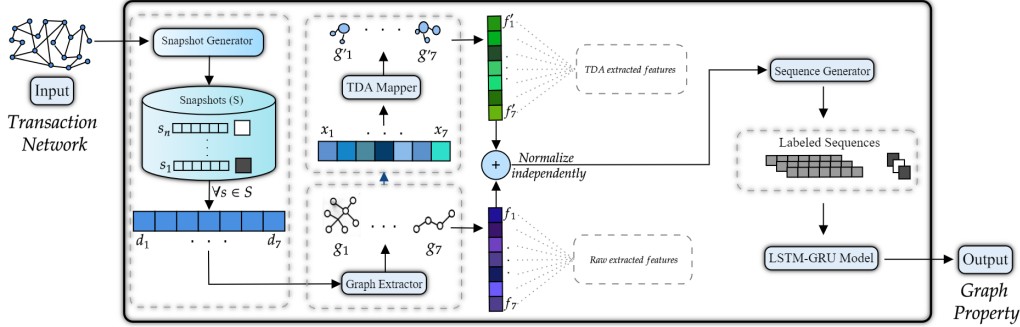

Figure 3: **GraphPulse Flowchart**. The GraphPulse system workflow starts with snapshot graph extraction, followed by the generation of Mapper networks. Next, a sequential LSTM+GRU model is developed, incorporating features from both the original snapshot graphs and the Mapper networks.

node features $\mathcal{X} \in \mathbb{R}^{N \times d}$ from the snapshot graph $\mathcal{G}_{t_i}$, which serve as inputs for the TDA Mapper (see Section 3). Example node features are the count of neighbors and the summation of incoming edge weights, details of which are given in Section 6. TDA Mapper creates $n$ Mapper networks for $n$ snapshots, which serve as the input for the sequential modeling step, as we explain next.

In the sequential modeling step, we aggregate pertinent features from both snapshot graphs and Mapper networks such as the number of nodes and edges (see Section A.3 for our selected features). Denoting snapshot graph features as $F_{\text{snapshot}}$ and Mapper network features as $F_{\text{Mapper}}$, where $F_{\text{snapshot}}$ includes graph-level characteristic features of the snapshot graph such as the number of nodes and edges, and $F_{\text{Mapper}}$ encompasses characteristics such as node count and average node size. The fusion of $F_{\text{snapshot}}$ and $F_{\text{Mapper}}$ provides a holistic perspective on the system's evolution. Equipped with these features, we compose a sequence spanning $n$ days, denoted as $S_n$, serving as input for our specialized sequential model. This sequential model illuminates the dynamic interplay between the changing graph property and the structural insights extracted from Mapper networks in an organized manner.

The specific nature of the predicted graph property dictates the form of the sequential model. For instance, for real-valued properties, a regression model may be appropriate, while binary properties can be addressed using a classification model. This adaptable approach aligns the model's structure with the unique characteristics of the targeted graph property.

## 6 EXPERIMENTS

**Graph property.** We use network growth in terms of edge count as the predicted graph property. Formally, let $\mathcal{G}$ be a graph, $t$ be a specific time, $\delta_1$ and $\delta_2$ be time intervals, and $E(t_1, t_n)$ denote the multi-set of edges between times $t_1$ and $t_n$. We define the graph property $P$ as:

$$P(\mathcal{G}, t_1, t_n, \delta_1, \delta_2) = \begin{cases} 1, & \text{if } |E(t_n + \delta_1, t_n + \delta_2)| > |E(t_1, t_n)| \\ 0, & \text{otherwise.} \end{cases}$$

Setting $n = 7$, $\delta_1 = 1$, and $\delta_2 = 7$, we establish a meaningful and practical graph property. This choice of parameters is valuable due to the application of 7-day predictions, which hold significance in both financial contexts, such as Ethereum asset networks (Kim et al., 2021), where they can guide financial decisions, and in the realm of social network infrastructure, like Reddit, where they aid in planning maintenance activities.

**Datasets.** We perform experiments on MathOverflow (Paranjape et al., 2017) and Reddit-Body (Kumar et al., 2018) datasets, and seven ERC20 token networks that we have extracted from the Ethereum blockchain. The ERC20 token networks, consisting of real transaction data from the beginning period of the Ethereum blockchain, provide valuable insights into the dynamics of digital assets. A summary of the statistics of these datasets is presented in Table 2.

**Mapper aspects.** Section 3 constructed Mapper networks based on node features $\mathcal{X}$, which we construct by extracting outgoing edge weight sum, incoming edge weight sum, outgoing edge count, and incoming edge count (see Section A.3 for feature definitions.) We used 2D-TSNE as our lens and

Table 1: ROC-AUC results for the graph property prediction task. The **bold** results represent the best methods for each dataset, and the underlined results represent the second-best methods.

| Dataset | GIN | TDA-GIN | EvolveGCN | GRUGCN | HTGN | GraphPulse |
|---|---|---|---|---|---|---|
| Adex | $0.4484_{\pm 0.0681}$ | $0.6089_{\pm 0.0574}$ | $0.7167_{\pm 0.1096}$ | $0.6843_{\pm 0.1594}$ | $\underline{0.7330}_{\pm 0.0849}$ | $\mathbf{0.8928}_{\pm 0.0022}$ |
| Bancor | $0.5895_{\pm 0.0514}$ | $0.5114_{\pm 0.0496}$ | $0.7931_{\pm 0.1773}$ | $\underline{0.8588}_{\pm 0.0190}$ | $0.7412_{\pm 0.0629}$ | $\mathbf{0.8722}_{\pm 0.0013}$ |
| Aragon | $0.3915_{\pm 0.0608}$ | $0.4648_{\pm 0.0499}$ | $\underline{0.7939}_{\pm 0.0875}$ | $0.7854_{\pm 0.0556}$ | $0.7781_{\pm 0.0508}$ | $\mathbf{0.8926}_{\pm 0.0035}$ |
| Dgd | $0.5748_{\pm 0.0163}$ | $0.5789_{\pm 0.0469}$ | $\underline{0.7460}_{\pm 0.0225}$ | $0.6704_{\pm 0.0557}$ | $0.6861_{\pm 0.0530}$ | $\mathbf{0.7804}_{\pm 0.0062}$ |
| Coindash | $0.5065_{\pm 0.0408}$ | $0.5015_{\pm 0.0278}$ | $0.7002_{\pm 0.0561}$ | $0.7321_{\pm 0.0399}$ | $\underline{0.7530}_{\pm 0.0348}$ | $\mathbf{0.7904}_{\pm 0.0037}$ |
| Iconomi | $0.6079_{\pm 0.0651}$ | $0.5158_{\pm 0.0651}$ | $\underline{0.8379}_{\pm 0.0327}$ | $0.8105_{\pm 0.0230}$ | $0.8221_{\pm 0.0139}$ | $\mathbf{0.8518}_{\pm 0.0044}$ |
| Centra | $0.4252_{\pm 0.0916}$ | $0.4777_{\pm 0.1042}$ | $0.8663_{\pm 0.1302}$ | $\underline{0.9001}_{\pm 0.0040}$ | $\mathbf{0.9044}_{\pm 0.0082}$ | $0.8670_{\pm 0.0077}$ |
| Reddit-Body | $0.3563_{\pm 0.0608}$ | $0.5281_{\pm 0.0777}$ | $\underline{0.7709}_{\pm 0.0667}$ | $0.6591_{\pm 0.0765}$ | $0.6557_{\pm 0.0401}$ | $\mathbf{0.8692}_{\pm 0.0070}$ |
| Mathoverflow | $0.3789_{\pm 0.0867}$ | $0.5953_{\pm 0.1045}$ | $0.6734_{\pm 0.0306}$ | $0.7405_{\pm 0.0938}$ | $\underline{0.7881}_{\pm 0.0509}$ | $\mathbf{0.8008}_{\pm 0.0050}$ |

KMeans for the clustering algorithm. We set the Mapper hyperparameters as $cls = 5$, $n\_cubes = 2$, and $perc\_overlap = 0.4$ and study their impact in Section A.4.

$F_{\mathbf{snapshot}}$ **and** $F_{\mathbf{Mapper}}$. In the sequential modeling step, we collect specific features from both snapshot graphs and Mapper networks. From the snapshot graphs, we extract three key features: the number of nodes, the number of edges, and the average value of edge weights. Additionally, we leverage Mapper networks to extract five supplementary features: the number of nodes, the number of edges, the maximum cluster size, the average cluster size, and the average value of edge weights.

**Models.** We compare GraphPulse against two baselines and three state-of-the-art TGNNs models (Appendix B details node, edge features, and graph types in models):

- The two baseline approaches adopt the powerful GIN framework (Xu et al., 2018) within a binary graph classification context. We feed a static graph into GIN, encompassing edges from days $t$ to $t + 7$, denoted as $\widehat{\mathcal{G}}^i = (\mathcal{G}^i_{t_1} \cup \mathcal{G}^i_{t_2} \cup \cdots \cup \mathcal{G}^i_{t_7})$ to predict the graph property. In a second variant of GIN, we integrate cluster membership information from Mapper networks as additional node features in a model called TDA-GIN. This baseline is designed to gauge the contribution of the information within Mapper networks.

- The state-of-art models in TGNNs are EvolvedGCN (Pareja et al., 2020), GRUGCN (Seo et al., 2018) and HTGN (Yang et al., 2021). EvolvedGCN focuses on evolving graph structures, GRUGCN incorporates variational principles, and HTGN handles discrete time intervals in temporal graphs. The models are notable advancements in the field of temporal GNNs.

**Implementation details.** GIN and TDA-GIN models use a Graph Isomorphism Network with 64 hidden units followed by a target output dimension of two. Raw RNN and TDA RNN models utilize LSTM and GRU layers with an Adam optimizer and a learning rate of $1 \times 10^{-4}$. A hybrid LSTM-GRU model processes sequences in a (7,3) and (7,5) format for input, respectively. We evaluate the models using the AUC-ROC score, a suitable metric for prediction assessment. We ran all experiments on a *Dell PowerEdge R630*, featuring an *Intel Xeon E5-2650 v3 Processor (10-cores, 2.30 GHz, 20MB Cache)*, and 192GB of RAM *(DDR4-2133MHz)*.

Table 2: Dataset statistics. $\overline{|\mathcal{E}_{\text{daily}}|}$ and $\overline{|\mathcal{V}_{\text{daily}}|}$ denote the average number of daily edges and nodes, respectively. $\overline{|\mathcal{G}|}$ indicates the total number of snapshots. *Reported as (years,months,days)*.

| **Dataset** | $\overline{|\mathcal{E}_{\mathbf{daily}}|}$ | $\overline{|\mathcal{V}_{\mathbf{daily}}|}$ | Duration* | $|\mathcal{G}|$ |
|---|---|---|---|---|
| Adex | 259.15 | 126.41 | (0,10,6) | 293 |
| Bancor | 320.84 | 154.97 | (0,10,24) | 311 |
| Aragon | 367.99 | 189.08 | (0,11,19) | 337 |
| Dgd | 90.16 | 29.32 | (2,0,7) | 720 |
| Coindash | 346.55 | 128.61 | (0,9,11) | 268 |
| Iconomi | 205.57 | 85.62 | (1,6,12) | 542 |
| Centra | 294.24 | 140.85 | (0,9,5) | 261 |
| Reddit-Body | 688.84 | 86 | (1,1,20) | 399 |
| Mathoverflow | 2532.75 | 124.09 | (0,6,16) | 183 |

## 6.1 EVALUATION RESULTS

Table 1 shows the ROC-AUC results for the two baselines (GIN, TDA-GIN) and three TGNN models. GIN has $> 0.5$ AUC only in four of the nine datasets. All results are averages of five runs. Incorporating topological information into GIN as node features, TDA-GIN improves the AUC val-

ues in five datasets. The largest gain is noted in the MathOverflow dataset with $+0.216$. However, the increased AUC values of TDA-GIN are still low.

The temporal GNN models have consistently high AUC values with HTGN having the highest mean AUC of $0.770$ across datasets. EvolvedGCN's mean AUC is $0.764$, while GruGCN follows closely with a mean AUC of $0.759$. GraphPulse has a mean AUC value of $0.849$ and has the highest AUC value for eight out of nine datasets.

GraphPulse employs features from both snapshot graphs and Mapper networks within a sequential model. This naturally raises the question: which specific features contribute to the predictive capability of GraphPulse? To address this question, we conduct an ablation study, where, we utilize graph features and Mapper network features in isolation within the same sequential model to predict the targeted graph property. Table 3 indicates that the median AUC value for $F_{\text{snapshot}}$-RNN is $0.7981$, while for $F_{\text{Mapper}}$-RNN, the value is $0.8501$. While Mapper-based features result in an overall higher AUC, it's noteworthy that Graph-Pulse achieves a considerably higher median AUC of $0.8670$, surpassing the AUC of both feature sets in isolation. This observation provides compelling evidence that the topological insights acquired through Mapper offer complementary information to the snapshot graph-based features.

Table 3: Ablation study. showcasing ROC-AUC values by incorporating graph features $F_{\text{snapshot}}$ and Mapper network features $F_{\text{Mapper}}$ within a sequence-based model.

| Data | $F_{\text{snapshot}}$-RNN | $F_{\text{Mapper}}$-RNN |
|---|---|---|
| Adex | $0.8673\pm 0.0027$ | $\mathbf{0.8831}\pm 0.0050$ |
| Bancor | $0.7981\pm 0.0078$ | $\mathbf{0.8501}\pm 0.0018$ |
| Aragon | $0.6898\pm 0.0794$ | $\mathbf{0.8819}\pm 0.0014$ |
| Dgd | $\mathbf{0.7689}\pm 0.0090$ | $0.7314\pm 0.0106$ |
| Coindash | $0.7676\pm 0.0025$ | $\mathbf{0.7790}\pm 0.0018$ |
| Iconomi | $0.8404\pm 0.0204$ | $\mathbf{0.8417}\pm 0.0048$ |
| Centra | $0.8610\pm 0.0065$ | $\mathbf{0.8673}\pm 0.0068$ |
| Reddit-Body | $\mathbf{0.8690}\pm 0.0070$ | $0.7735\pm 0.0125$ |
| Mathoverflow | $\mathbf{0.7798}\pm 0.0133$ | $0.7522\pm 0.0057$ |

To assess the importance of individual features within the $F_{\text{Mapper}}$-RNN, we conduct a secondary ablation study within the same sequence model for the Aragon network which has the largest number of nodes and edges per day among our datasets. This study aims to identify key features that significantly influence the $F_{\text{Mapper}}$-RNN model's performance. Table 4 indicates that removing the number of edges (shared nodes between Mapper clusters) causes the biggest drop in AUC values for this dataset. Additionally, we assessed the effectiveness of GraphPulse through two supplementary graph property prediction tests: density growth and node count growth. The detailed results for these tests are presented in Appendix D

Table 4: Ablation study. ROC-AUC values when removing a Mapper network feature from $F_{\text{Mapper}}$ within a sequence-based model for the Aragon network.

| Removed feature | ROC-AUC |
|---|---|
| Number of nodes | $0.8716\pm 0.0087$ |
| Number of edges | $0.7341\pm 0.0466$ |
| Max cluster size | $0.8799\pm 0.0024$ |
| Average cluster size | $0.8650\pm 0.0089$ |
| Average edge weight | $0.8530\pm 0.0409$ |

**Scalability.** Due to space limitations, we report the computational complexity and scalability results in Appendix E. Here we note that on the largest token network, GraphPulse completes the training process 26% faster than the time required by the state-of-the-art HTGN method. Furthermore, TDA Mapper can process snapshot graphs of 20,000 nodes in under 4 minutes.

## 7 CONCLUSION

We have introduced GraphPulse, a principled approach for predicting graph properties in temporal graphs. By leveraging a combination of snapshot graphs and Mapper networks, GraphPulse capitalizes on both structural and topological insights to enhance prediction accuracy. Through empirical evaluations, we have demonstrated the effectiveness of GraphPulse across diverse datasets, showcasing its superior performance compared to baseline methods. Notably, our approach demonstrates scalability across both training and analysis phases, making it an adaptable solution for large-scale temporal graph scenarios. GraphPulse presents a promising advancement in the field of temporal graph property prediction, bridging the gap between structural and topological aspects for accurate predictions.

## REPRODUCIBILITY

In line with ICLR's commitment to promoting reproducibility in research, we have taken diligent steps to ensure the reproducibility of our work presented in this paper. We provide open access to our source code, which is available at the following link: `https://github.com/kiarashamsi/GraphPulse`. This code includes the implementation of our novel models and algorithms, ensuring that fellow researchers can readily access and reproduce our experimental results.

## ETHICS STATEMENT

We would like to emphasize our commitment to conducting ethical research throughout the course of this study. This research primarily involves the analysis and prediction of temporal graph properties using machine learning techniques. We have taken ethical considerations into account at various stages of our work. Our experiments do not involve human subjects, and we have not accessed any sensitive or private information. The datasets used in this study are publicly available, and for the Ethereum token networks analyzed, the raw data is publicly accessible to anyone on the web. We do not share any address or transaction labels for the token networks. Our chosen graph property, network growth, does not involve any individual address.

## ACKNOWLEDGEMENTS

This work was partially supported by the Canadian NSERC Discovery Grant RGPIN-2020-05665: **Data Science on Blockchains**, the National Science Foundation grants DMS-2202584 and DMS-2220613, and the Simons Foundation grant # 579977. This research was also supported by the Canadian Institute for Advanced Research (CIFAR AI chair program). Shenyang Huang is supported by Natural Sciences and Engineering Research Council of Canada (NSERC) Postgraduate ScholarshipDoctoral (PGS D) Award and Fonds de recherche du Québec - Nature et Technologies (FRQNT) Doctoral Award.

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

# Appendix

## A   TDA MAPPER & GRAPH TRAJECTORY

**TDA Mapper Example.** We give a toy example for TDA Mapper networks in Figure 4 where for the point cloud $\mathcal{X}$ we use the height function $f : \mathcal{X} \to \mathbb{R}$ as a lens. We use 6 intervals $\mathcal{I} = \{I_1, \ldots, I_6\}$ to cover $f(\mathcal{X})$ (Figure 4b). By using the chosen clustering algorithm, we detect the clusters in each $f^{-1}(I_k)$ for each $k$ (Figure 4a). These clusters form elements of a cover of $\mathcal{X}$. In particular, as illustrated in Figure 4a, $f^{-1}(I_2)$ induces two clusters of points that are enclosed by the blue rectangles, while $f^{-1}(I_1)$ produces a single cluster enclosed by an orange rectangle. The Mapper network in Figure 4c is the nerve of this cover, where each node represents the corresponding cluster and each edge represents that the clusters have non-empty intersections. The final Mapper network gives a rough summary/sketch of the whole point cloud (Hajij et al., 2018). In TDA Mapper graphs, nodes are associated with clusters of data points, and edges are drawn between nodes based on the overlap or commonality of data points between the clusters. The edge weight reflects the strength of this connection, indicating how many data points are included in the overlapped area.

### A.1   MAPPER TRAJECTORIES

Let $G$ be a graph where a monotonically increasing or decreasing function $f$, such as the number of unique neighbors, is defined over the nodes, indicating changes by the addition of new nodes or edges. Assume that the addition of a new node or edge pair creates a graphlet whose isomorphic copies exist in $G$. Furthermore, assume that the function $f$ remains unchanged over the nodes of $G'$, a modified graph where node features are retained, and over $N'$, the nodes of the new graphlet, which have features similar to those in the existing nodes $N$ (or the changes are for few nodes only and minimal, as in a star-shaped graph and new nodes to be added at the periphery). If Mapper is used to create a 2D representation that assigns data points to specific cubes, the following holds true:

—The addition of $N'$ does not change the position of $N$ within the Mapper network. Specifically, the cluster that contains $N$ can include $N'$ without any modification.

**Proof:** By construction, Mapper identifies clusters based on similar function values. Since $N'$ and $N$ have similar function values, they fall into the same cluster within the Mapper network. The addition of $N'$ does not affect the existing cluster structure since it shares the same function values as $N$.

— Consequently, the graphlet's addition to $G$ does not change the similarity of $G$ to $G'$.

**Proof:** Since $N$ and $N'$ fall into the same cluster within the Mapper network, this implies that their topological features are similar in $G$ and $G'$. Therefore, the addition of the graphlet, which did not change the features of $N$, does not alter the similarity between $G$ and $G'$. In other words, $\text{sim}(G, G')$ remains unchanged.

—The graphlet that is added to $G$ only changes $\text{sim}(G, G')$ if it brings additional information to the graph.

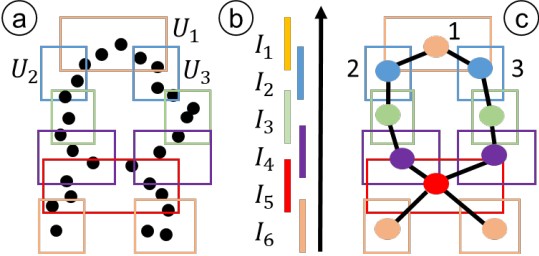

Figure 4: **Toy Example of Mapper.** For a point cloud $\mathcal{X}$ (a), we define a lens function $f : \mathcal{X} \to \mathbb{R}$ (b), and the induced covering defines a Mapper network where nodes represent clusters and edges represent related clusters (c).

**Proof:** By the definition of graph similarity, $\text{sim}(G, G')$ quantifies the amount of information shared between the two graphs. If the graphlet introduced into $G$ does not bring any new information or topological changes to $G$, then $\text{sim}(G, G')$ remains the same. It only changes if the graphlet contributes new, distinct features or structure to $G$.

Kolmogorov complexity Li et al. (2008) supports our argument. Kolmogorov complexity is a measure of the shortest possible length of a program that can generate a particular piece of data. In the context of our argument, if the graphlet does not change the structure or features of $G$, then the shortest program to generate $G$ and $G'$ remains the same. This indicates that the additional information brought by the graphlet can be quantified as the difference in program length, further supporting our argument.

## A.2 EMPIRICAL EVALUATION FOR MAPPER TRAJECTORIES

To demonstrate the efficiency of TDA Mapper in capturing trajectory (see Figure 8) information within the phase space, we implement the following experimental setup. Initially, we establish a reference graph and construct a grid of neighboring points positioned within the phase space around this reference graph (refer to Figure 6). Each data point within this grid represents a graph instance generated using the specific parameters corresponding to its location in the phase space.

Formally, we define $n^{th}$-*neighbourhood* (or $n$-shell) of a graph $\mathcal{G}_0$ in the phase space $\mathcal{P}$ as the set of all graphs in $\mathcal{P}$ whose distance to $\mathcal{G}_0$ is exactly equal to $n$ with respect to supremum norm. For example, in Figure 5, the red dot represents the reference graph $\mathcal{G}_0$, while all light blue graphs represent $1^{st}$-neighbourhood of $\mathcal{G}_0$.

For instance, the reference graph can be an Erdős-Rényi graph characterized by $p = 0.5$ and $n = 50$. Its phase space neighbors are produced by increasing and decreasing the values of $n$ and $p$, resulting in a graph with the chosen parameter pair. Consider Figure 5 as an example; an immediate neighbor, *neighbor 7*, is generated with $p = 0.5$ and $n = 51$.

Within this grid, we establish two key criteria. Firstly, reference graph should exhibit greater similarity to its 1-hop neighbors compared to those beyond 1-hop. Secondly, the 1-hop neighbors should possess similarity values that are nearly identical since they are equidistant from the reference graph.

To this end, we choose *Erdős-Rényi graphs* (Erdős & Rényi, 1960) for their widespread usage and *Stochastic Block Models* (Holland et al., 1983) for their ability to capture community structures within networks. We experimented on a sample of 250 reference graphs generated by the Erdős-Rényi graph generator where we varied the number of nodes $n$ from 30 to 50 and $p$ from 0.21 to 0.4 to make the phase space. For the Barabási-Albert model, the phase base is defined with the number of nodes

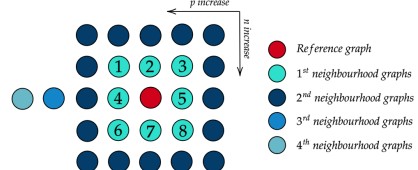

Figure 5: **Erdős-Rényi Graph neighbourhoods.** The red dot indicates the reference graph, with its neighbourhood graphs in the phase space. first neighbourhood indexed from 1 to 8.

$n$ from 120 to 500 and the number of edges to attach from a new node to existing nodes $m$ from 20 to 100. We randomly choose the value for each parameter for each model 50 times and use that combination of parameters to generate reference graphs five times.

**Graph Similarity Measure.**

There are numerous graph similarity measures (Zager & Verghese, 2008). In our experiments, we use *the eigenvalue method* for graph similarity measure (Koutra et al., 2011), where graph similarity is defined based on the eigenvalues of the graph Laplacian. We chose this method because it allows us to compare networks of different sizes, which is ideal for temporal graphs where nodes and edges vary across time snapshots. The eigenvalue method involves utilizing the eigenvalues of the graph Laplacian, and it proves to be suitable for our specific scenario.

Let $\mathcal{A}_1$ and $\mathcal{A}_2$ be two adjacency matrices of graphs $\mathcal{G}_1$ and $\mathcal{G}_2$, respectively. Let $\mathcal{L}_1 = \mathcal{D}_1 - \mathcal{A}_1$ and $\mathcal{L}_2 = \mathcal{D}_2 - \mathcal{A}_2$ be the Laplacians of the graphs, where $\mathcal{D}_1$ and $\mathcal{D}_2$ are the corresponding diagonal matrices of degrees. Considering that the eigenvalues of $\mathcal{L}_j$ are denoted as $\{\lambda_{j1}, \lambda_{j2}, ...\}$, we define

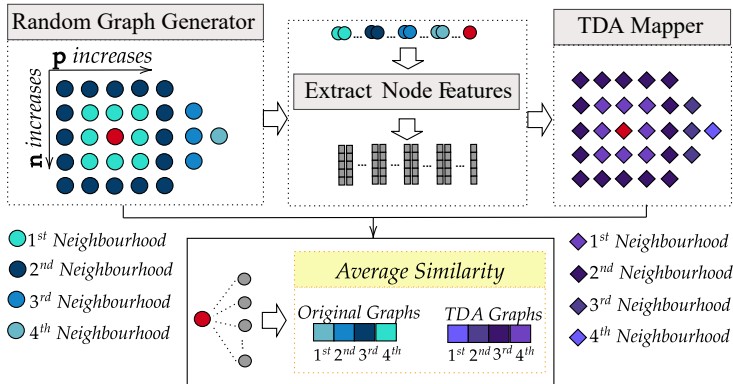

Figure 6: **Simulating Graph Similarity**. We generate $k$ graphs using a graph generator model (e.g., Erdős-Rényi) with parameters ($p$ and $n$). Average similarity to neighbors in phase spaces is computed, revealing insights into graph continuity across parameter values.

the similarity between the graphs as follows:

$$\mathbf{sim}(\mathcal{G}_1, \mathcal{G}_2) = \sum_{i=1}^{k} (\lambda_{1i} - \lambda_{2i})^2$$

where $k$ is the smallest value that satisfies the following condition:

$$\min\{\frac{\sum_{i=1}^{k} \lambda_{1i}}{\sum_{i=1}^{|\mathcal{V}_1|} \lambda_{1i}}, \frac{\sum_{i=1}^{k} \lambda_{2i}}{\sum_{i=1}^{|\mathcal{V}_2|} \lambda_{2i}}\} > 0.9$$

It is understood as the top $k$ eigenvalues containing 90% of the energy Koutra et al. (2011). Please note that this method yields unbounded similarity scores within the $[0, \infty]$ range. When the dissimilarity score approaches 0, it indicates a high degree of dissimilarity between the graphs, whereas higher values suggest greater dissimilarity (Koutra et al., 2011).

**Similarity Score Comparison.** Having selected a similarity function, we now turn our attention to how we compare the Mapper trajectories with those of the snapshot graphs. Figure 7 illustrates the main idea in our experiments for comparing the similarity scores of original graphs and their topological counterparts.

First, for any graph $\mathcal{G}$, we use TDA Mapper to induce a summary graph $\Gamma$ (which is the Mapper network of $\mathcal{G}$) by using its node features. We consider the following features to represent the nodes: PageRank, Degree Centrality, Closeness Centrality, Betweenness Centrality, clustering coefficient, and the number of neighbors. We consider these features because they can be computed from any graph without node or edge labels, which is preferable in our setting where labels might be difficult

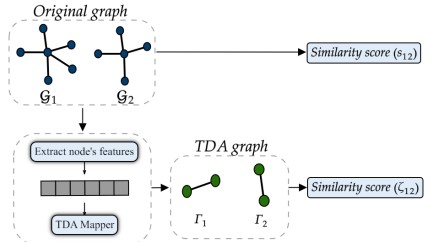

Figure 7: **Similarity score comparison.** Using original graphs and extracted TDA Mapper graphs based on original node features for similarity score comparison.

to collect. For each graph $\mathcal{G}$, the nodes $\mathcal{V}$ are represented as a point cloud $\mathcal{X}_{\mathcal{G}}$ in the feature space $\mathbb{R}^{N \times d}$. Then, by applying TDA Mapper on the point cloud $\mathcal{X}_{\mathcal{G}} \subset \mathbb{R}^{N \times d}$, we obtain its Mapper network $\Gamma_{\mathcal{G}}$.

Now, consider two neighboring graphs $\mathcal{G}_1$ and $\mathcal{G}_2$ in the phase space (e.g., Erdős-Rényi, Barabási-Albert), and let their similarity score be $s_{12}$, i.e. $s_{12} = \mathrm{sim}(\mathcal{G}_1, \mathcal{G}_2)$. Now, let $\Gamma_1$ and $\Gamma_2$ be their induced Mapper networks. Let $\zeta_{12}$ be their similarity score, i.e., $\zeta_{12} = \mathrm{sim}(\Gamma_1, \Gamma_2)$ By our definition, if $\zeta_{12}$ is smaller than $s_{12}$, then the induced Mapper networks are more similar to the original graphs. This interprets that Mapper networks inherit the similarity information better than the original graphs, hence they capture graph trajectory better.

**Results.** In Tables 5 and 6, we report the median similarity scores for original graphs and TDA Mapper networks in $\log_{10}$ base for both Erdős-Rényi and Barabási-Albert setting. Due to the unstable

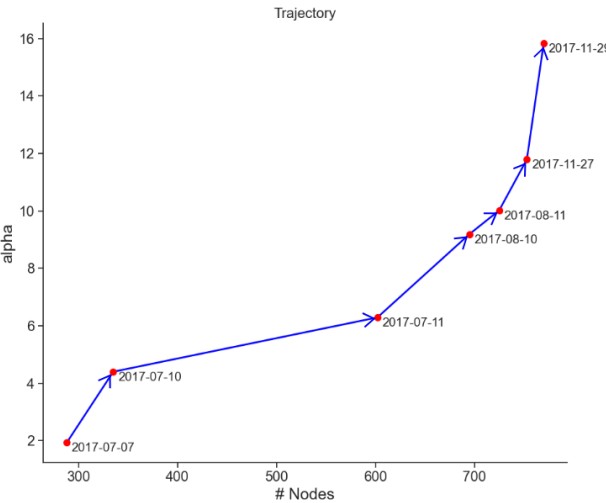

Figure 8: **Adex Network Trajectory.** A seven-day trajectory for the Adex network within the phase space of $\alpha$ and $|V|$. We assume that the network follows a power law model (Adamic & Huberman, 2000), and fit graph data to compute the $\alpha$ exponent of the model (see appendix C). For the network, the power law exponent $\alpha$ moves from 2 to 16.

frequency of outliers, we use the median to calculate the average similarity score of four neighbour-hoods instead of the mean since the median is more robust to outliers than the mean. However, both Mapper networks and original graphs become increasingly dissimilar to the reference graph as the distance (modeled with the k-hop neighbourhood) in the phase space increases. Considering this, we see a significant increase in the similarity score in the original graphs while the induced TDA Mapper networks remains highly steady.

Table 5: Dissimilarity scores in the Erdős-Rényi setting with their standard deviations.

| Neighborhood | TDA ($log_{10}$) | Original ($log_{10}$) | $\sigma_{TDA}$ | $\sigma_{original}$ |
|---|---|---|---|---|
| 1 | 0.9364 | 2.0726 | 0.15 | 0.25 |
| 2 | 0.9430 | 2.6166 | 0.11 | 0.23 |
| 3 | 0.9548 | 2.9734 | 0.09 | 0.22 |
| 4 | 0.9769 | 3.2091 | 0.08 | 0.22 |

Table 6: Dissimilarity scores in the Barabási-Albert setting with their standard deviations.

| Neighborhood | TDA ($log_{10}$) | Original ($log_{10}$) | $\sigma_{TDA}$ | $\sigma_{original}$ |
|---|---|---|---|---|
| 1 | 0.9392 | 3.6165 | 0.33 | 0.20 |
| 2 | 0.9516 | 4.0612 | 0.31 | 0.19 |
| 3 | 0.9756 | 4.3289 | 0.32 | 0.20 |
| 4 | 0.9797 | 4.5013 | 0.30 | 0.20 |

Figure 9 shows that TDA Mapper yields nearly identical dissimilarity values for the neighbors of the reference graph. The result further offers evidence that TDA Mapper based trajectories would better capture the graph's moves in the phase space.

With the two criteria fulfilled by these empirical results, we conclude that the TDA Mapper can efficiently capture and model graph trajectories in the phase space.

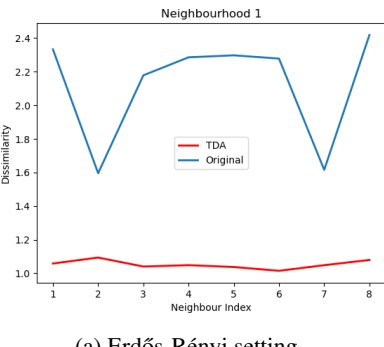 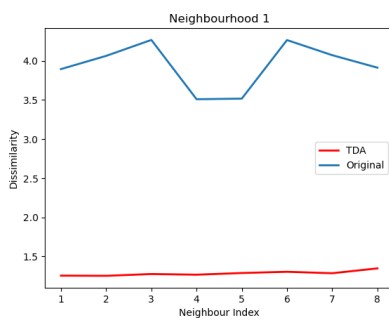

(a) Erdős-Rényi setting.  (b) Barabási-Albert setting.

Figure 9: **Initial Neighbourhood Comparison.** Comparison of similarity scores in the initial neighbourhood for Erdős-Rényi and Barabási-Albert configurations. Refer to Figure 5 for neighbor indices. Given the proximity of these neighbors to the reference graph in phase space, we expect their dissimilarity to exhibit both low variability and similarity among themselves. Notably, TDA Mapper scores demonstrate greater stability and lower values than the dissimilarity scores of the original graphs.

**Stability of Mapper networks.** We also want to underline the stability of Mapper networks with respect to small changes, and in particular, their robustness against noise. As Figure 5 shows every reference graph (red dot) has 8 neighbors (light blue dots) in their first neighbourhood in a 2-parameter phase space (e.g., Erdős-Rényi, Barabási-Albert). In Figures 9a and 9b, we consider the first neighbourhoods of the reference graphs, and give the average similarity scores between the reference graphs and these 8 graphs in their first neighbourhood for both Erdős-Rényi, Barabási-Albert settings. Changes in the Erdős-Rényi and Barabási-Albert parameters result in fluctuating the dissimilarity score between the original and reference graphs. Each time a new neighbor graph is created, the number of nodes $n$ and probability $p$ increases or decreases by 2 and 0.05, respectively. These small changes are considered as noise to the graph's trajectory. While there is a fluctuation in dissimilarity scores between the original graph, dissimilarity scores between TDA graphs are stable. Therefore, it is concluded that the TDA method is robust to the noise of graph trajectory.

## A.3 FEATURES FOR TDA MAPPER

To induce our TDA Mapper networks for a given graph $\mathcal{G}$, we use the following node features to induce a point cloud $\mathcal{X}_\mathcal{G}$ in the feature space $\mathbb{R}^N$ and follow the method described in Section 3. The node features we use are as follows:

1. **Outgoing Edge Weight Sum:** For each node in the snapshot graph, this feature calculates the sum of the weights of all outgoing edges connected to that node. It provides information about the total influence or importance of the node in sending information to its neighbors.

2. **Incoming Edge Weight Sum:** Similar to the previous feature, this one calculates the sum of the weights of all incoming edges connected to each node in the snapshot graph. It represents the total influence or importance of the node in receiving information from its neighbors.

3. **Outgoing Edge Count:** This feature keeps track of how many edges leave each node in the snapshot graph. The number of other nodes the node is directly connected to as well as its level of connectedness are reflected by this.

4. **Incoming Edge Count:** The number of incoming edges to each node in the snapshot graph is counted by this feature. It gives details on how many nodes are connected to a single node directly.

## A.4 DEPENDENCY ON MAPPER PARAMETERS

The configurability of TDA Mapper is enhanced by its set of hyperparameters. These parameters provide users with the ability to customize TDA Mapper's functionality, allowing for alignment with

Table 7: Similarity scores in the Erdős-Rényi setting with "rough" Mapper parameters, which constitute an *imposed failure* scenario.

| neighborhood | TDA ($log_{10}$) | Original ($log_{10}$) | $\sigma_{TDA}$ | $\sigma_{original}$ |
|---|---|---|---|---|
| 1 | 0.0 | 2.0743 | 0.0 | 0.28 |
| 2 | 0.0 | 2.5988 | 0.0 | 0.25 |
| 3 | 0.0 | 2.9810 | 0.0 | 0.24 |
| 4 | 0.0 | 3.2105 | 0.0 | 0.25 |

their specific data and analytical goals. This adaptability facilitates the comprehensive capture and visualization of topological features within diverse datasets. The hyperparameters of TDA Mapper include:

- **Number of Cubes** ($n\_cubes$): Number of hypercubes along each dimension of the projected point cloud using the lens function.

- **Percentage of Overlaps** ($perc\_overlap$): Percentage of overlap between adjacent cubes calculated along each dimension.

- **Number of Clusters** ($cls$): Number of clusters in the K-means algorithm that determines the number of inner clusters formed within each cube.

The effectiveness of the GraphPulse can be influenced by the choice of parameters in the Mapper algorithm.

In experiments conducted in A.2, we set the hyperparameters as $cls = 5$, $n\_cubes = 2$, and $perc\_overlap\ (gain) = 0.4$ to induce our Mapper networks, where *cls* represents how fine the clustering in the point cloud, $n\_cubes$ (interval size) represents how big the clusters are, and finally, $perc\_overlap$ represents how fine the connections between the clusters are. Our experimental results in Figure 10 show that Mapper network representations remain stable under small changes in the phase space.

It is noteworthy that the level of view granularity in Mapper is an important parameter. If we increase $perc\_overlap$ significantly, this will add an edge between most Mapper clusters, and the resulting Mapper network will be a very dense (sometimes complete) graph. We show this dependency with the following experiment in the following Mapper

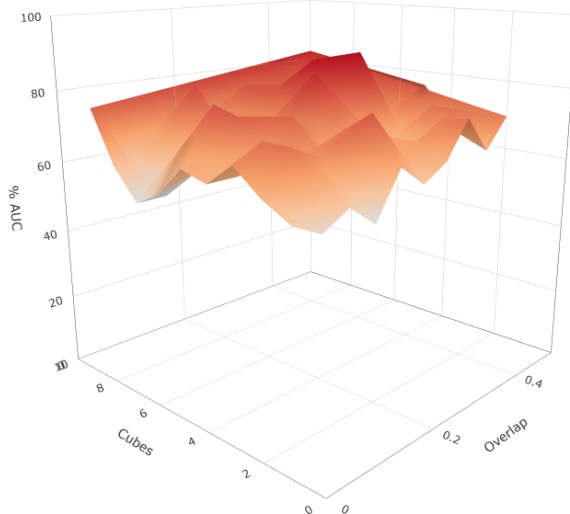

Figure 10: **Adex Network Mapper Analysis.** 3d AUC plot over Mapper parameter for the Adex network. Predictive performance (ROC-AUC) is consistently above 0.8 for the region $0.2 < overlap < 0.45$ and $4 < cubes < 8$.

setting: $cls = 5$, $n\_cubes = 2$, and $perc\_overlap = 0.7$. In Table 7, we see that the resulting TDA graphs are summarizing too much, and there is no change in the Mapper network even if the original graphs are changing.

Table 8: Summary of analyzed models.

| Model | Type | Data | Edge Feature | Node Feature |
|---|---|---|---|---|
| GIN | GNN | $\{\mathcal{G}_{t_1} \cup \mathcal{G}_{t_2} \cup \cdots \cup \mathcal{G}_{t_n}\}$ | edge weight | — |
| TDA-GIN | GNN | $\{\mathcal{G}_{t_1} \cup \mathcal{G}_{t_2} \cup \cdots \cup \mathcal{G}_{t_n}\}$ | edge weight | mapper cluster membership |
| EvolvedGCN | TGNN | $\mathcal{G}_{t_1}, \mathcal{G}_{t_2} \dots \mathcal{G}_{t_n}$ | edge weight | — |
| GRUGCN | TGNN | $\mathcal{G}_{t_1}, \mathcal{G}_{t_2} \dots \mathcal{G}_{t_n}$ | edge weight | — |
| HTGN | TGNN | $\mathcal{G}_{t_1}, \mathcal{G}_{t_2} \dots \mathcal{G}_{t_n}$ | edge weight | — |
| $F_{\text{mapper}}$-RNN | TDA+RNN | $\mathbf{\Gamma}_1, \dots, \mathbf{\Gamma}_n$ | edge weight | NA |
| $F_{\text{snapshot}}$-RNN | RNN | $\mathcal{G}_{t_1}, \mathcal{G}_{t_2} \dots \mathcal{G}_{t_n}$ | edge weight | NA |
| GraphPulse | TDA+RNN | $\mathcal{G}_{t_1}, \mathcal{G}_{t_2} \dots \mathcal{G}_{t_n}$ | edge weight | — |

# B  BASELINE MODELS

In our evaluation of different models, we included baseline models using Graph Neural Networks (GNNs) and Recurrent Neural Networks (RNNs) to tackle our graph-based property prediction task. Here, we will provide explanations for all the baseline models employed in our study. Table 8 presents an overview of the model summary information.

## B.1  GNNs AND OUR MODELS

**GNNs:** For the GNN baselines, we employed the Graph Isomorphism Network (GIN) as a static model due to its remarkable capacity for capturing both local and global graph structures, making it adaptable for tasks such as graph classification, node classification, and link prediction. Also, we used three state-of-the-art GNN models including EvolvedGCN, GRUGCN, and HTGN as baselines.

**GIN.** We extract each snapshot graph and augment it with four essential node features. The following features are included: Outgoing Edge Weight Sum, Incoming Edge Weight Sum, Outgoing Edge Count, and Incoming Edge Count.

After incorporating these four node features into the graph representation, Based on the chronological order, the graphs are divided into 80% training and 20% testing data, which are then fed into the model. We employ a Graph Isomorphism Network (GIN) model for graph classification. The GIN model consists of four middle layers with 64 hidden units followed by a target output dimension of two, which serves as label prediction. We use Adam optimizer with a learning rate of 0.0001.

**TDA-GIN.** The TDA-GIN method is a two-step approach that builds upon the previous static GIN method. First, we extract the same graph with the four node features explained earlier for each node. Additionally, we leverage these node features as input for a TDA Mapper algorithm, which forms clusters by grouping similar nodes together. This process yields a new TDA graph, and for each node in this graph, we incorporate the cluster size as a node feature.

We perform a grid search on parameter combinations to determine the ideal combination of TDA Mapper parameters. We choose the most advantageous combination through this optimization process in order to produce the most instructive TDA graph for learning and classification. The TDA-GIN method enhances the representation and classification of temporal graphs by incorporating TDA techniques into the GIN framework, leading to better prediction performance.

While TDA-GIN incorporates additional structural information through the TDA Mapper representation, it has been observed that for temporal tasks, this method may not always yield improved results. The temporal nature of the data brings challenges related to dynamic changes, time dependencies, and evolving patterns, which may not be fully captured by the TDA-GIN approach. As a result, the benefits gained from TDA-based graph representations may not always translate into superior performance for temporal graph property prediction tasks.

**RNNs:** For our RNN baseline, we developed a hybrid LSTM-GRU model, combining the strengths of both architectures to address the specific challenges posed by our dataset and task. A hybrid LSTM+GRU model often exhibits superior performance compared to standalone LSTM and GRU

models due to its ability to effectively combine the unique advantages of both architectures. LSTM excels at capturing long-range dependencies in sequential data, making it suitable for tasks involving context over extended sequences. On the other hand, GRU is computationally more efficient and can capture short-term dependencies effectively (Yamak et al., 2019). By blending these two architectures into a hybrid model, we harness the capacity to capture both short and long-term dependencies simultaneously. This enables the model to better understand the complex temporal relationships present in the data, which might be challenging for standalone LSTM or GRU models to grasp individually. Moreover, A hybrid LSTM+GRU model leverages the diverse internal structures and regularization mechanisms of LSTM and GRU models to improve performance and generalization in sequential data tasks, making it a versatile choice. This combination creates an ensemble-like effect, enhancing the model's ability to capture different data features and reduce overfitting risk.

$F_{\textbf{snapshot}}$**-RNN.** The $F_{\text{snapshot}}$-RNN method entails building a temporal daily pipeline to capture the temporal component of the data. We extract daily graphs for each day from each snapshot, which contains seven days' worth of data. We extract three graph-level features from each graph, including the number of nodes, the number of edges, and the average value of edge weights. We give this feature a constant value if the graph is unweighted. After these features are extracted, a sequence of three features is created for seven consecutive days, producing a sequence with the shape (7,3) for each snapshot. To carry out the classification task, a hybrid LSTM-GRU model gets these sequences and their corresponding labels. Based on the chronological order, the sequences are divided into 80% training and 20% testing data, which are then fed into the model. Our RNN model consists of two LSTM layers, two GRU layers, and a dense layer for classification. Each snapshot has binary classification labels in the model's output. The AUC-ROC is the evaluation metric employed for this task.

$F_{\textbf{Mapper}}$**-RNN.** The $F_{\text{Mapper}}$-RNN method consists of two main steps. In the first step, similar to the previous model, we construct daily graphs from the temporal data. For each daily graph, we utilize the four features previously discussed in the initial section to generate a TDA graph. This involves extracting the four features for each node in the daily graphs and subsequently constructing a TDA Mapper network for each individual day.

From the TDA daily graphs, we extract five additional features, namely the number of nodes, the number of edges, the maximum cluster size, the average cluster size, and the average value of edge weights with the shape (7,5) for each snapshot. These features are then used to create a sequence representing seven consecutive days. This sequence is fed into the same LSTM-GRU model described before for the classification task. The subsequent steps are identical to the previous model. The sequences are partitioned into 80% training and 20% testing data, based on the chronological order, and are employed as input for the hybrid LSTM-GRU model. The model outputs binary classification labels for each snapshot, and the performance is evaluated using the AUC-ROC metric.

### B.2 Temporal Graph Representation Learning Methods

Our work relates to graph representation learning in a dynamic setting. Here, we further elaborate temporal graph learning methods considered as baselines and provide the experimental details.

**EvolveGCN** (Pareja et al., 2020) captures the dynamism of a graph sequence, EvolveGCN adapts a GCN architecture and uses an RNN to evolve the GCN parameters through the temporal aspect. In this way, the weights of the GCN are dynamically and automatically updated by the RNN module allowing the GCN module to change even during the test time.

**GCRN** (or GRUGCN) (Seo et al., 2018) is a deep learning model aiming to predict structured sequences of data. GCRN basically exploits a CNN architecture and applies it to the graph data to identify the spatial structures. Furthermore, it utilizes an RNN module on top of that to encompass the dynamic of the network.

**HTGN** (Yang et al., 2021). While most temporal graph representation learning methods focus on modeling structural and temporal dependencies in Euclidean space, Hyperbolic Temporal Graph Networks (HTGN) grasp the high capacity and hierarchical awareness of hyperbolic space. HTGN utilizes hyperbolic graph neural networks and hyperbolic gated recurrent neural networks to capture the evolutionary patterns of dynamic graphs.

It is noteworthy that these methods are mainly optimized for node-level tasks namely node classification. Since we are focusing on the graph property prediction task, we modify the decoder component of these methods. In particular, we added a pooling layer on top of the encoder module to provide a graph-level representation. The output of the pooling layer is then fed to the final classifier for the downstream task of graph property prediction.

***Experimental Details.*** The experimental parameters are set according to the best practices proposed in baseline methods (Yang et al., 2021). For all these methods, we used the in-degree, out-degree, weighted in-degree, and weighted out-degree of the nodes as their initial features. We set the final embedding dimension as 16, and used a *mean*-pooling layer for generating graph-level representations. All methods are composed of one layer of recurrent units and two-layer graph convolutions. For HTGN, the number of historical windows in the HTA module is set to 5. For all methods, we utilized a chronological %80–%20 split of the graph snapshot sequence as our *train-validation* and *test* data, respectively.

## C    THE PHASE SPACE FOR TOKEN NETWORKS

A token network on Ethereum can be conceptualized as a dynamic graph-based decentralized ecosystem. In this ecosystem, nodes represent participants or entities within the network, which can include users, smart contracts, or even devices. Edges between nodes symbolize various interactions and transactions involving tokens. These interactions could include transfers, trades, or other token-related activities.

We model token networks as power law graphs Adamic & Huberman (2000) where the phase space is given with $|V|$ and the exponent $\alpha$. In power law graphs, $P(x) \propto x^{-\alpha}$ where x is the node degree. Table 9 shows the $\alpha$ values, as described by Alstott et al. (2014), for token networks. A high alpha indicates that there are fewer nodes with very high degrees (hubs) compared to nodes with lower degrees. This leads to a more concentrated distribution, where a small number of nodes have extremely high degrees, and the majority of nodes have relatively low degrees. Different real-world systems exhibit different alpha values, reflecting the diversity in

Table 9: $\alpha$ values for the power-law distributions of the degree distributions of 7 token datasets

| Data | $\alpha$ |
| --- | --- |
| Adex | 4.4387 |
| Bancor | 3.8886 |
| Aragon | 3.3336 |
| Dgd | 3.3696 |
| Coindash | 4.8846 |
| Iconomi | 3.3369 |
| Centra | 3.7404 |

the distribution of node degrees. For instance, in the World Wide Web, alpha values typically range from 2 to 3 Adamic & Huberman (2000). As the table shows, the token networks exhibit significantly high $\alpha$ values.

## D    ADDITIONAL GRAPH PROPERTIES

In addition to network growth in edges, we have carried out two experiments to test the predictive power of GraphPulse: predicting the network growth in node count and density.

Table 10 displays the results of the node count experiments. Among the nine datasets utilized in the experiment, GraphPulse achieves the highest AUC in five datasets and the second-highest AUC in the Iconomi dataset. EvolveGCN attains the highest AUC value in the four datasets; however, it produces poor results in Adex and Coindash. Among the methods, GraphPulse boasts the highest mean AUC (0.8044). HTGN consistently delivers high AUC results, consequently achieving the second-highest mean AUC at 0.7912.

Table 11 illustrates the results for the density property prediction experiments. In this task, Graph-Pulse outperforms other methods in four of nine datasets and is the second-best method in the Bancor and Aragon datasets. EvolveGCN attains the highest AUC value in three datasets; however, it produces poor results in Coindash and Bancor. HTGN and GRUGCN both achieve the best result in only one dataset. Among the methods, HTGN has the best mean AUC (0.7558). GraphPulse has

the second-best mean AUC at (0.7403) however GraphPulse has the best median value (0.7832) and consistently delivers high AUC results for all of the datasets with AUC higher than (0.65)

Table 10: ROC-AUC results for the graph node count prediction task.

| Dataset | GIN | TDA-GIN | EvolveGCN | GRUGCN | HTGN | GraphPulse |
|---|---|---|---|---|---|---|
| Adex | $0.5899_{\pm0.0486}$ | $0.5002_{\pm0.0180}$ | $0.5699_{\pm0.3690}$ | $0.5566_{\pm0.3141}$ | $\underline{0.7720}_{\pm0.1100}$ | $\mathbf{0.8224}_{\pm0.0047}$ |
| Bancor | $0.4724_{\pm0.0377}$ | $0.5879_{\pm0.0627}$ | $0.8078_{\pm0.1688}$ | $\underline{0.8165}_{\pm0.0266}$ | $0.6859_{\pm0.0781}$ | $\mathbf{0.8182}_{\pm0.0115}$ |
| Aragon | $0.5121_{\pm0.0918}$ | $0.4880_{\pm0.0096}$ | $\underline{0.7020}_{\pm0.0886}$ | $0.6637_{\pm0.0380}$ | $0.6335_{\pm0.0223}$ | $\mathbf{0.7416}_{\pm0.0116}$ |
| Dgd | $0.5334_{\pm0.0519}$ | $0.5400_{\pm0.0625}$ | $\mathbf{0.8291}_{\pm0.0609}$ | $0.7497_{\pm0.0629}$ | $\underline{0.8115}_{\pm0.0263}$ | $0.7851_{\pm0.0046}$ |
| Coindash | $0.4582_{\pm0.0965}$ | $0.4637_{\pm0.0497}$ | $0.6055_{\pm0.1954}$ | $0.7900_{\pm0.019}$ | $\underline{0.7969}_{\pm0.0194}$ | $\mathbf{0.8078}_{\pm0.0067}$ |
| Iconomi | $0.6515_{\pm0.0200}$ | $0.5512_{\pm0.0335}$ | $\mathbf{0.9086}_{\pm0.0240}$ | $0.8505_{\pm0.0200}$ | $0.8334_{\pm0.0261}$ | $\underline{0.8582}_{\pm0.0069}$ |
| Centra | $0.5034_{\pm0.0994}$ | $0.5637_{\pm0.0248}$ | $0.8221_{\pm0.1334}$ | $0.8726_{\pm0.0035}$ | $\underline{0.8753}_{\pm0.0040}$ | $\mathbf{0.8790}_{\pm0.0046}$ |
| Reddit-B | $0.4995_{\pm0.0329}$ | $0.5365_{\pm0.0411}$ | $\mathbf{0.9408}_{\pm0.0164}$ | $0.8147_{\pm0.0516}$ | $\underline{0.8289}_{\pm0.0424}$ | $0.7283_{\pm0.0092}$ |
| Mathoverflow | $0.6314_{\pm0.0632}$ | $0.4559_{\pm0.0784}$ | $\mathbf{0.9257}_{\pm0.0026}$ | $0.8352_{\pm0.0485}$ | $\underline{0.8933}_{\pm0.0530}$ | $0.8404_{\pm0.0041}$ |

Table 11: ROC-AUC results for the graph density prediction task.

| Dataset | GIN | TDA-GIN | EvolveGCN | GRUGCN | HTGN | GraphPulse |
|---|---|---|---|---|---|---|
| Adex | $0.4763_{\pm0.0645}$ | $0.4929_{\pm0.0206}$ | $\underline{0.7634}_{\pm0.3037}$ | $0.7234_{\pm0.3040}$ | $0.7356_{\pm0.2419}$ | $\mathbf{0.7938}_{\pm0.0017}$ |
| Bancor | $0.5320_{\pm0.0367}$ | $0.5529_{\pm0.0209}$ | $0.5810_{\pm0.1683}$ | $\mathbf{0.8120}_{\pm0.0154}$ | $0.7492_{\pm0.0753}$ | $\underline{0.7661}_{\pm0.0077}$ |
| Aragon | $0.4571_{\pm0.1195}$ | $0.4896_{\pm0.0514}$ | $\mathbf{0.7938}_{\pm0.0887}$ | $0.6419_{\pm0.0151}$ | $0.6680_{\pm0.0511}$ | $\underline{0.7832}_{\pm0.0050}$ |
| Dgd | $0.5228_{\pm0.0755}$ | $0.5475_{\pm0.0581}$ | $\mathbf{0.8711}_{\pm0.0292}$ | $0.7881_{\pm0.0274}$ | $\underline{0.8171}_{\pm0.0108}$ | $0.7958_{\pm0.0088}$ |
| Coindash | $0.4326_{\pm0.0646}$ | $0.4694_{\pm0.0406}$ | $0.5021_{\pm0.1506}$ | $\underline{0.7744}_{\pm0.0051}$ | $0.7679_{\pm0.0043}$ | $\mathbf{0.7767}_{\pm0.0125}$ |
| Iconomi | $0.7209_{\pm0.0582}$ | $0.5086_{\pm0.0588}$ | $\mathbf{0.9146}_{\pm0.0082}$ | $\underline{0.8927}_{\pm0.0085}$ | $0.8784_{\pm0.0051}$ | $0.8444_{\pm0.0105}$ |
| Centra | $0.3991_{\pm0.0417}$ | $0.5622_{\pm0.0746}$ | $0.7196_{\pm0.1631}$ | $0.8434_{\pm0.0028}$ | $\underline{0.8491}_{\pm0.0017}$ | $\mathbf{0.8908}_{\pm0.0050}$ |
| Reddit-Body | $0.4014_{\pm0.0569}$ | $0.5329_{\pm0.0236}$ | $\underline{0.7072}_{\pm0.0437}$ | $0.6055_{\pm0.0633}$ | $0.6093_{\pm0.0347}$ | $\mathbf{0.7270}_{\pm0.0109}$ |
| Mathoverflow | $\underline{0.7421}_{\pm0.0469}$ | $0.5133_{\pm0.0710}$ | $0.6720_{\pm0.1883}$ | $0.7335_{\pm0.0352}$ | $\mathbf{0.7475}_{\pm0.0284}$ | $0.6846_{\pm0.0100}$ |

## E  SCALABILITY ANALYSIS

The computational cost of GraphPulse is dominated by the cost of reducing our 4D feature data $\mathcal{X}$ to a $2D$ form to be used by Mapper. We have used tSNE (Van der Maaten & Hinton, 2008) for the reduction which has a quadratic computational complexity in the number of data points. The cost can be significantly reduced by using tSNE approximations (Pezzotti et al., 2016). We leave this improvement to future work. Once a lens is selected, creating the TDA Mapper network involves sorting data points (graphs), which is an $O(n\,log n)$ operation where $n$ is the number of graphs.

We demonstrate GraphPulse's scalability through two key aspects: end-to-end model training costs for the most resource-intensive dataset, and the computational overhead of Mapper analysis for daily snapshot graphs.

Figure 11 illustrates the computational time requirements of the considered models on the Dgd network, which boasts the largest number

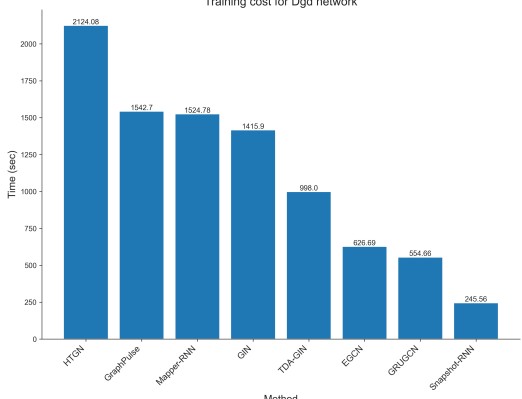

Figure 11: **Methods Training Time.** Comparison of Training Time for Methods on the Dgd Network. GraphPulse completes the training process 26% faster than the time required by the state-of-the-art HTGN method.

of snapshots (720). Notably, GraphPulse completes training in *1550* seconds, while HTGN demands over *2100* seconds for the same task.

The processing time of GraphPulse encompasses three distinct stages: the extraction of TDA sequences, the extraction of daily sequences, and the training of the RNN model. Notably, the TDA

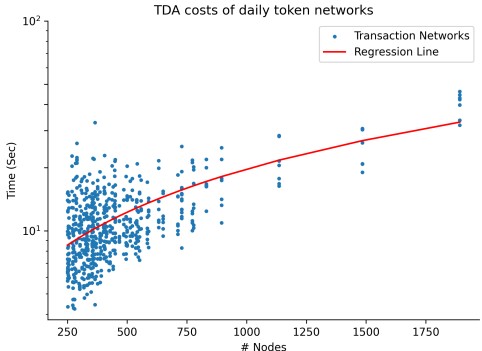

(a) Daily cost of TDA Mapper creation for all token networks. Each data point is a snapshot graph from a token. As shown, most of the daily graphs have less than 750 nodes.

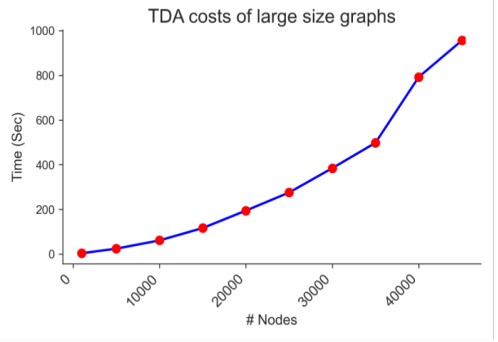

(b) The cost of extracting TDA features for graphs using the Erdős-Rényi graph generation model with $p = 0.3$.

Figure 12: **TDA Mapper Scalability.** The daily cost of TDA Mapper for token networks is illustrated in (a). The mapper cost has a direct relation with the size of the graph. (b) shows the increasing cost of TDA Mapper with the growing size of the graph.

sequence extraction phase is the most time-consuming component of this process. Consequently, the overall execution time for GraphPulse and $F_{\text{Mapper}}$-RNN is significantly higher when compared to the $F_{\text{Snapshot}}$-RNN model, primarily due to the additional computational requirements of TDA sequence extraction.

Given that the extraction of TDA sequences constitutes a significant portion of GraphPulse's processing time, we conducted a comprehensive analysis focused on the daily TDA cost. Figure 12 presents the outcomes of this analysis, shedding light on the question of how substantial the daily cost can be while still permitting efficient TDA processing. Remarkably, the results illustrate that TDA remains highly effective, even with graphs containing approximately 20,000 nodes, processing them in under 4 minutes. Furthermore, the experiments reveal that a majority of the daily networks processed exhibit fewer than 1,000 nodes, ensuring swift processing. This evidence underscores the scalability of TDA for real-world daily graphs within the GraphPulse framework.

These results collectively underscore GraphPulse's outstanding scalability in temporal graph machine learning. By efficiently managing training costs and Mapper analysis, GraphPulse offers a high degree of scalability across various datasets, establishing its suitability for larger and more complex temporal graph scenarios.

