# OpenReview forum: "GraphPulse: Topological representations for temporal graph property prediction"
_ICLR.cc/2024/Conference — ICLR 2024 poster_

### Official Review · Reviewer_yFgn · 2023-10-30

**Soundness:** 3 good
**Presentation:** 2 fair
**Contribution:** 3 good
**Rating:** 8
**Confidence:** 3

**Summary:**

The paper proposes an approach (GraphPulse) for learning on Discrete Time Dynamic Graphs (DTDGs). GraphPulse relies on the mapper method (a Topological Data Analysis technique used to compress high dimensional data in a small and comprehensible graphical representation) to extract latent topological properties of each graph snapshot. More in detail, provided a sequence of graphs (G_1, G_2, …, G_N), for each graph G_i, the authors extract a finite set of “snapshot” features describing the general properties of G_i (i.e. the
number of nodes, the number of edges, and the average value of edge weights). They then proceed at computing for G_i a compact graphical representation using the mapper method, and from the obtained graph they extract 5 new “mapper” features (the number of nodes, the number of edges, the maximum cluster size, the average cluster size, and the average value of edge weights). The snapshot and mapper features of each graph are then fed in input to a sequential model (LSTM+GRU), which outputs a prediction for the sequence of graphs. To evaluate their method, the authors defined a network growth problem where they aim at predicting whether a given network in a future interval will show more edges than in the past or not.

**Strengths:**

The paper illustrates an interesting application of a Topological Data Analysis technique for constructing a model able to tackle prediction tasks on DTDGs. While the approach is not quite straightforward to explain due to the multiple steps that are involved, the paper is generally clear (see although some weaknesses on this below). Experimental evaluation on a variety of datasets show good performance of the method for the considered task.

**Weaknesses:**

While I found the manuscript rather interesting and generally well presented, in some parts I had some difficulties completely understanding the paper. For instance, in Section 3, the clustering method is not really introduced in the mapper method and I found it difficult at first to understand the role that this would have in the overall approach. Similarly, in the experimental section I did not find details on what clustering algorithm or lens function the authors used for computing the Mapper Network (edge weights for the mapper network and mapper hyperparameters in Section 6 are also not defined in the paper). I would greatly appreciate it if the authors could provide some details in their rebuttal on this to clarify my understanding of their paper (and in general improve readability).

For what concerns the experimental evaluation, the results look promising, however I wonder if using a GIN as a static GNN baseline might not be fair. The goal of the model is to predict whether in a future window of time there will be more edges in the graph than in the past. As such, being able to understand whether the network is growing in size (or not) over time seems a priori a meaningful feature for the model. This information can however not be captured with a GIN, as the only features this has access to are: Outgoing EdgeWeight Sum, Incoming EdgeWeight Sum, Outgoing Edge Count, and Incoming Edge Count; which do not describe such evolution. For this reason I wonder if decorating edge features with the timestamp associated to the time when an edge appears and using an architecture able to compute graph embeddings based on edge features (e.g. a MPNN) could be a better baseline for the given prediction task.

**Questions:**

I found it interesting that the authors handcrafted a feature descriptor for both a snapshot and the mapper network. While reading the paper I was expecting to see a GNN for extrapolating graph-wise features that would have then been used in input to the recurrent model. This would have provided (ignoring the construction of the mapper network with a fixed lens function) an end-to-end learnable approach that would have indeed suited a generic classification problem. I wonder if the authors have experimented with such a solution in their experiments and whether this could provide a further boost in performance.

---

> ### Author Response · Authors · 2023-11-17
> **Author response to reviewer yFgn**
>
> We thank the reviewer for their time and valuable feedback.
>
> **W1 I would greatly appreciate it if the authors could provide some details in their rebuttal on this to clarify my understanding of their paper (and in general improve readability)**
>
> **A:**  Thank you for your valuable feedback. We have addressed your comments by adding parameter definitions to the Appendix A.4. We added further clarification for our lens and clustering methods in Section 6 of the main paper. Moreover we included a definition for edge weight in TDA graphs in Appendix A. Your feedback helped us further improve the clarity of our paper.  We will also address your comments below.
> For the lens function, we are using a 2D-TSNE and for the clustering algorithm, we used the KMeans algorithm for clustering the points inside each cubes. Also regarding the hyperparameters definition of the TDA mapper, our mapper includes three hyperparameters:
>
> * **Number of Cubes (n_cubes):** Number of hypercubes along each dimension of the projected point cloud using the lens function.
>
> * **Percentage of Overlaps (perc_overlap):** Percentage of overlap between adjacent cubes calculated along each dimension.
>
> * **Number of Clusters (cls)**: Number of clusters in the K-means algorithm that determines the number of inner clusters formed within each cube.
>
> Finally, In TDA Mapper graphs, nodes are associated with clusters of data points, and edges are drawn between nodes based on the overlap or commonality of data points between the clusters. The edge weight reflects the strength of this connection, indicating how many data points are included in the overlapped area.
>
> **W2  I wonder if decorating edge features with the timestamp associated to the time when an edge appears and using an architecture able to compute graph embeddings based on edge features (e.g. a MPNN) could be a better baseline for the given prediction task**
>
> **A:** Thank you for the review. In fact, static baselines are a good first check on what graph neural networks can achieve but we do not expect them to yield SOTA results in temporal graph tasks. For this reason, we employ three well-known SOTA methods in our experiments and show that GraphPulse outperforms all of them.
> In response to your suggestion, we conducted a new experiment for the time stamped Adex dataset. We introduced the time (day) of edge appearance as an additional edge feature in our GIN model. Indeed, as you suggested, these modifications led to notable improvements in the static GIN results. Overall, we still observe that our GraphPulse method and other temporal baselines outperforms static GIN as expected.
>
> | Model                   | Result  |
> |-------------------------|---------|
> | Static GIN                               | 0.44842 |
> | GIN with Temp. Edge Feat           | 0.61304 |
> | HTGN                    | 0.73308 |
> | GRUGCN                  | 0.6843  |
> | EGCN                    | 0.71678 |
> | GraphPulse (Ours)              | 0.8928  |
>
> **Q1 I wonder if the authors have experimented with such a solution in their experiments and whether this could provide a further boost in performance**
>
> **A:** Thank you for your suggestion. Indeed, designing an end-to-end learnable approach is an interesting idea and your suggestion is employed by the EvolveGCN method, which is a baseline compared in our experiments.  The input to EvolveGCN is a sequence of graph snapshots and it captures the network's dynamic by using an RNN to update the GCN parameters over time. Here, for each timestamp, all the node embeddings are aggregated (via mean pooling) to generate a graph-level embedding and then EvolveGCN is adapted to train end-to-end for the temporal graph property prediction task. Empirically, we show that our GraphPulse method consistently outperforms EvolveGCN. We plan to explore the possibility of applying Mapper directly to the graph without relying on node features as a future direction for an end-to-end approach based on GraphPulse.

---

> > ### Comment · Reviewer_yFgn · 2023-11-17
> >
> > I thank the reviewer for their detailed response. I just have a follow up question in light of their rebuttal. With the new static baseline you added the day of appearance as an edge feature in GIN. However, the GIN model as defined in the original paper doesn't include the use of edge features in the definition of the convolutional layer. How have those edge features been used in the model? did you simply combine them to the original node features of the neighbours and added zero padding to the target node or you used an additional MLP on the neighbors features (before summation) for dimensionality reduction? I would be personally in favour of the latter to enrich the quality of the representation and provide the fairest comparison (the former approach would not necessarily produce an injective function over the neighbourhood, which in turn would not necessarily produce a maximally expressive GNN). Also I think that the analysis should be run with all the datasets of Table 1, not just Adex.

---

> > > ### Author Response · Authors · 2023-11-21
> > > **Author response to reviewer yFgn**
> > >
> > > Thank you for your suggestions and the inspiring review; we have completed your requested experiments for all the networks for the *network growth* prediction task.
> > > Initially, we presented the static GIN as a baseline for our task and compared it with temporal SOTA GNN models. Based on your review we then developed the GIN+Temporal Edge Feature model as an improved model over the base static GIN. Next, we also added the MLP layer that you have requested. The results of these three models are presented in the table below.
> > > The GIN model with Temporal Edge Feature improved the results of static GIN in 6 out of 9 datasets. Next, using the MLP layer with GIN+Temporal Edge Feature also improved the AUC results of GIN  in 6 out of 9 cases. Finally Using the MLP+Temp.Edge Feature improves the AUC of 7 datasets over the GIN+Temp Edge Feature without MLP.  The improved results of the model with the MPL layer show the importance of this layer; furthermore, in two networks (i.e., Adex and Dgd) the model performed slightly better than SOTA GNN models. However GraphPulse yields better AUC values than all GIN variations. Thanks again for sharing your suggestion.
> > >
> > >
> > >
> > > |                         | Adex  | Bancor | Aragon | Dgd   | Coindash | Iconomi | Centra | Reddit-Body | Mathoverflow |
> > > |-------------------------|-------|--------|--------|-------|----------|---------|--------|-------------|--------------|
> > > | Static GIN              | 0.4484| 0.5895 | 0.3915 | 0.5748| 0.5065   | 0.6079  | 0.4252 | 0.3563      | 0.3789       |
> > > | GIN with Temporal Edge Feat | 0.613 | 0.5304 | 0.3431 | 0.5964| 0.649    | 0.4536  | 0.4412 | 0.5582      | 0.46         |
> > > | GIN with Temporal Edge Feat + MLP| 0.7685 | 0.4884 | 0.5355 | 0.6917 | 0.6759 | 0.575   | 0.6778 | 0.5433      | 0.5153       |
> > > | HTGN                    | 0.733 | 0.7412 | 0.7781 | 0.6861| 0.753    | 0.8221  | 0.9044 | 0.6557      | 0.7881       |
> > > | GRUGCN                  | 0.6843| 0.8588 | 0.7854 | 0.6704| 0.7321   | 0.8105  | 0.9001 | 0.6591      | 0.7405       |
> > > | EGCN                    | 0.7167| 0.7931 | 0.7939 | 0.746 | 0.7002   | 0.8379  | 0.8663 | 0.7709      | 0.6734       |
> > > | GraphPulse(Ours)        | 0.8928| 0.8722 | 0.8926 | 0.7804| 0.7904   | 0.8518  | 0.867  | 0.8692      | 0.8008       |

---

> ### Comment · Reviewer_yFgn · 2023-11-21
>
> Thank you for your response, that clarifies my last doubt. In light of the review and the additional experiments of Tables 10 and 11 the authors provide, I'm willing to raise my score to acceptance

---

> > ### Author Response · Authors · 2023-11-21
> > **Author response to reviewer yFgn**
> >
> > We express our gratitude for your thoughtful review and for taking the time to engage in the discussion period and raise the score. Your suggestions helped us improve our work significantly.

---

### Official Review · Reviewer_Qyn2 · 2023-10-31

**Soundness:** 3 good
**Presentation:** 4 excellent
**Contribution:** 3 good
**Rating:** 8
**Confidence:** 5

**Summary:**

This paper introduces a new framework called GraphPulse for predicting the evolution of temporal graphs, combining topological data analysis and recurrent neural networks. One of the most important insights of this study is that the evolution of a graph can be represented as a temporal trajectory in a Newtonian phase space. This advances the understanding of the principle of graph evolution for this research community. The proposed GraphPulse is evaluated on financial and cryptocurrency transaction networks and compared to state-of-the-art methods, showing a significant performance improvement.

**Strengths:**

1. This paper identified that the evolution of a graph can be represented as a temporal trajectory in a Newtonian phase space and conducted comprehensive studies on it. Existing study [1] has pointed out that the evolution of a dynamic graph can be treated as a trajectory in a latent space without further studying what space it is and any properties the space should have. This study advances the understanding of the principle of graph evolution, which I believe is very valuable to this research community.
[1] Time-Capturing Dynamic Graph Embedding for Temporal Linkage Evolution, TKDE.
2. A novel temporal graph embedding framework is proposed, which is highly effective at capturing the evolution of temporal graphs in the phase space. The proposed GraphPulse is technically sound.
3. GraphPulse demonstrates significant performance improvement compared to state-of-the-art dynamic graph neural networks.
4. The authors create cryptoasset networks for the temporal graph property task and publish them as temporal benchmark datasets for future research.

**Weaknesses:**

1. It is unclear how hyperparameters sensitive the model is.
2. Since GraphPulse is a generic framework applicable to extend the embedding algorithms to embed a temporal graph, the generalizability of GraphPulse should be discussed and tested. It is suggested to replace Mapper with two or more static graph embedding techniques to test the impact on GraphPulse.
3. There are some typos. For example, in page 18, there are two different “neighborhood” styles of writing.

**Questions:**

Is the GraphPulse applicable in link prediction, edge sign prediction in signed networks, edge/node attribute prediction? Compared to the embedding algorithms that are customized for those particular graph mining applications, what are the advantage of the proposed GraphPulse? It would be better to discuss the application conditions of GraphPulse so that the audience knows under what application conditions GraphPulse will get better performance.

---

> ### Author Response · Authors · 2023-11-17
> **Author response to reviewer Qyn2**
>
> We thank the reviewer for their time and valuable feedback.
>
> **W1 It is unclear how hyperparameters sensitive the model is.**
>
> **A:** Thanks for raising this point. Due to the page limit for the main text, a comprehensive hyperparameter sensitivity analysis was included in Appendix A.4. The key finding from this analysis is that within a well-defined and extensive region within the hyperparameter space,  GraphPulse consistently achieves the best performances, with an AUC exceeding 0.7 as illustrated in Figure 10.
>
> **W2 Since GraphPulse is a generic framework applicable to extend the embedding algorithms to embed a temporal graph, the generalizability of GraphPulse should be discussed and tested**
>
> **A:** We thank the reviewer for raising this question. We believe that the topological information extracted from Mapper is a key component to GraphPulse as seen in experiments (Section 6) and using Mapper in GraphPulse is an important architectural choice. In terms of generalizability, GraphPulse can generalize to multiple graph properties such as growth, density, node count, and more, and achieve SOTA performance on all of them. We also tested GraphPulse extensively on nine networks from two distinct domains and GraphPulse is able to generalize to all of them with strong performance on each. Indeed as the reviewer suggested, GraphPulse is a generic framework that could incorporate other embedding algorithms and future work can utilize additional embedding algorithms.
>
> **W3 There are some typos. For example, in page 18, there are two different “neighborhood” styles of writing**
>
> **A:** Thank you for the review. We will consolidate our writing and remove one of the conflicting styles. We have fixed the neighborhood issue in the paper.
>
> **Q1 Is the GraphPulse applicable in link prediction, edge sign prediction in signed networks, edge/node attribute prediction? Compared to the embedding algorithms that are customized for those particular graph mining applications, what are the advantage of the proposed GraphPulse?**
>
> **A:** We thank the reviewer for discussing this point. We design the GraphPulse framework to generate graph level embeddings for the temporal graph property prediction task. Edge level tasks such as link prediction and edge sign prediction are not directly applicable for GraphPulse. However, the topological features captured by TDA also provide node embeddings which can be beneficial to node and edge level tasks.
>
> The real advantage of GraphPulse is that it represents graphs in a Newtonian phase space, as illustrated in Figure 8. By constructing the phase space, we can treat the graphs as if they were particles in a dynamic system. This enables us to predict the trajectory of the graph, i.e., where the graph will be at the next time step, based on its current state. Based on the trajectory, we can predict new nodes (in Figure 2a, the first two graphs where a node is added), we can predict motifs (Figure 2a, the third graph is developing triangles and can be expected to get denser), we can identify node features (Figure 2a, leaf nodes co-appearing in the mapper cluster having similar features). GraphPulse allows for customized applications due to the fact that the mapper tool is highly customizable, and once the mapper network is created, we can represent graphs, node subsets or edge subsets in a phase space to be used by GraphPulse.

---

> > ### Comment · Reviewer_Qyn2 · 2023-11-20
> >
> > I would like to thank the authors for their detailed responses. I appreciate that representing the graphs in the Newtonian phase space and treating their evolutions as the trajectory path in the space is promising and novel. The authors also indicate a very important point that the mapper tool is highly customizable for representing graphs and node/edge subsets. I am wondering how to customize the GraphPulse so that it can be well applied to node/edge subsets or widely graph mining applications. It is worth discussing in the paper so that the followers can have a better judgment about whether they should directly use GraphPulse or develop a customized model when building their own solutions to solve similar problems in the future.
> >
> > As reviewer NvJb mentioned, Mapper cannot well capture the connection information. Although GraphPulse can effectively capture a global view of graph evolution based on simple node statistics and node features, which shows good results in the experiments, I believe Mapper is not the only model that can achieve this goal as GraphPulse is a general framework. Could you please discuss why Mapper is the best choice for GraphPulse? Doesn't changing Mapper to something incorporating the connection information get better performance?

---

> > > ### Author Response · Authors · 2023-11-21
> > > **Author response to reviewer Qyn2**
> > >
> > > We thank the reviewer for the discussion.
> > >
> > > In this work, Mapper graphs are actually generated based on graph structural information.As shown in  Figure 3 GraphPulse Flow Chart, the input to the mapper are node features which are based on the structural encodings of each node in the graph. In this work, we refer to these node structure encodings as node features which are detailed in Appendix A.3.
> > >
> > > The node structural encodings include the sum of outgoing/incoming edge weight, outgoing/incoming edge count which encode the local connectivity information regarding each node. The Mapper summaries are then produced based on these structural encodings which have the following distinct advantages (detailed in Section 4.3).
> > >
> > > * First, mapper networks capture **topological features**; intuitively, clusters in the Mapper network reflect the structural role of nodes in the graph. Figure 2 provides one visual example of this: going from G1 to G2 merely involves the addition of 2 satellite nodes (nodes that are only connected to the central hub node) this is reflected in the Mapper network as having 2 clusters and the cluster corresponding to satellite nodes increase from 3 nodes to 5 nodes (mirroring the change in the original graph). However, when going from G2 to G3, triangles are formed thus creating one additional cluster.
> > >
> > > * The second advantage of the Mapper network is its compressed representation. Mapper summaries capture a global view of interactions between nodes with different structural neighborhoods in the graph and demonstrate how the neighborhood structure evolves by leveraging the cluster sizes in Mapper when encoding the graph into an embedding.
> > >
> > > * Lastly, Mapper is multi-resolution as its cluster can correspond to nodes with various degrees.
> > >
> > > Considering all of the above advantages, we chose Mapper to capture the evolution of the global graph structure from the extracted node structural encodings. GraphPulse demonstrated strong performance over the baselines on the temporal graph property prediction task, which showcased the appealing capability of Mapper encodings.

---

> > > > ### Comment · Reviewer_Qyn2 · 2023-11-21
> > > >
> > > > Thank you for the response that addressed all my concerns. I suggest including the following in the amending manuscripts.
> > > > 1. The real advantage of GraphPulse in the response to Q1.
> > > > 2. The generalizability of GraphPulse in the response to W2.
> > > > 3. The distinct advantages of Mapper graphs.
> > > > 4. How GraphPulse advances existing literature on the understanding of the principle of graph evolution.
> > > >
> > > > In light of the authors' responses and the amendments mentioned above, I'm willing to raise my score to 8. Good job.

---

> > > > > ### Author Response · Authors · 2023-11-21
> > > > > **Author response to reviewer Qyn2**
> > > > >
> > > > > We would like to extend our sincere appreciation for your diligent review of our paper. Your insights and suggestions have been invaluable in improving our work. Based on your suggestions, we will elaborate and clarify on the aforementioned points suggested by you in the later manuscript revisions.

---

### Official Review · Reviewer_NvJb · 2023-11-02

**Soundness:** 2 fair
**Presentation:** 3 good
**Contribution:** 3 good
**Rating:** 6
**Confidence:** 5

**Summary:**

This paper studies the problem of temporal graph learning. The main contribution is GraphPulse, which combines structural and topological insights to predict graph properties. Experiments on ER graphs and SBM graphs demonstrate the superiority of the proposed method.

**Strengths:**

S1. The problem is well motivated by the need of tracking evolving graphs.

S2. It’s novel to incorporate the topological technical with dynamic graph represent learning. Using TDA Mapper to learn and predict graph trajectory is interesting.

**Weaknesses:**

W1. The authors use the set of nodes as the input point cloud of the Mapper network, aggregating the summation of incoming/outcoming edge weights and the counts of incoming/outcoming edges to conduct node features. However, there is a notable absence of retained graph structure information when compared to the original graph structure. In my opinion, the overall structure and the connections between nodes are more crucial and directly effective in the process of learning dynamic graphs. I would suggest the authors to incorporate the connection information into the model.

W2. The current assessment of the graph property focuses on determining whether there is a rise or decline in the number of edges, resulting in a fairly limited perspective. I would suggest the authors utilize more comprehensive graph properties to evaluate model capabilities on temporal graphs, such as changing node counts, density, diameter, degree distribution, and the number of triangles. Additionally, rather than binary classification, predicting the specific number of increased/decreased edges would test a finer-grained understanding of edge dynamics.

**Questions:**

See W1-W2 for details.

===== After rebuttal =====
I would like to thank the authors for answering precisely and comprehensively to my concerns. The extent of work and the additional experiments presented during this period are noteworthy. These efforts have enhanced my understanding of the model's principles, leading me to a more favorable evaluation. Consequently, I have raised my score to 6.

---

> ### Author Response · Authors · 2023-11-17
> **Author response to reviewer Nvjb**
>
> We thank the reviewer for their time and valuable feedback.
>
> **W1 I would suggest the authors to incorporate the connection information into the model**
>
> **A:** We thank the reviewer for raising this point. In this work, we focus on the graph property prediction task which requires a global understanding of the graph structure. Empirically, we find that the combination of Mapper representation and aggregated node features outperforms SOTA temporal GNNs based on local message passing. This might be because GraphPulse can effectively capture a global view of graph evolution based on simple node statistics and node features. In the literature, it is known that classical message passing based GNNs can suffer from various issues such as limited expressiveness (up to 1-WL test), over-squashing, and over-smoothing. How to best capture the global graph structure for GNNs remains one of the open research directions. Thus in this work, our proposed GraphPulse acts as a first approach for temporal graph property prediction and we leave it to future work to further improve upon GraphPulse by further encoding the global graph structure.
>
> **W2 Utilize more comprehensive graph properties to evaluate model capabilities on temporal graphs, such as changing node counts, density, diameter, degree distribution, and the number of triangles**
>
> **A:** We express our gratitude to the reviewer for bringing up this point. In this work, as a first approach, we formulate the temporal graph property prediction task as a binary classification task indicating the increase/decrease of graph properties in the future. This formulation can directly be applied to important applications such as deciding if the number of transactions in a cryptocurrency network will grow in the future or if the number of users will increase in a social network. As the reviewer pointed out, the next step would be to predict the extent of growth and decrease as a regression task which is left as an important future work due to the fact that regression tasks are much less studied in graph representation learning literature in general as compared to classification tasks and they pose additional challenges; for example, the magnitude of different properties can vary significantly over time.
>
> We also thank the reviewer for suggesting additional graph properties for evaluation. In this rebuttal, we have experimented with two additional properties, namely density and node counts (as kindly suggested by the reviewer). The detailed results are reported in Table 10 and Table 11 in Appendix D and we observe that GraphPulse performs the best overall. This suggests that the GraphPulse framework can generalize to a variety of graph properties.

---

> > ### Author Response · Authors · 2023-11-21
> > **Author response to reviewer Nvjb**
> >
> > We thank the reviewer for their detailed review. As the end of the discussion period is fast approaching, we would appreciate your feedback on our rebuttal.

---

### Official Review · Reviewer_ehy4 · 2023-11-03

**Soundness:** 3 good
**Presentation:** 4 excellent
**Contribution:** 2 fair
**Rating:** 6
**Confidence:** 2

**Summary:**

GraphPulse is a novel framework proposed for analyzing and predicting the evolution of temporal graphs through a combination of temporal graph neural networks and topological data analysis (TDA). The paper presents a process that takes snapshots of temporal graphs at fixed intervals, constructs TDA Mapper representations for these snapshots, and then uses these topological features along with snapshot graph features for sequential modeling to predict future graph properties. The paper claims superior performance over existing models in predicting network growth on several datasets, including new cryptocurrency network datasets.

**Strengths:**

1. The paper pioneers the integration of TDA into the study of temporal graphs, which represents a significant methodological advancement, potentially unlocking new insights into graph structure and evolution over time.

2. The paper is commended for its clear presentation style and the provision of solid supporting materials, which aid in the understanding and reproducibility of the proposed approach. The introduction of TDA Mapper method and the examples given are very helpful for audiences who do not have prior knowledges on TDA.

3. The proposed algorithm's simplicity and ease of implementation make it accessible for broad application across various temporal graph analysis tasks, facilitating its adoption in practice.

**Weaknesses:**

1. Focusing solely on the growth rate as a graph property is a limitation; the model's adaptability to other graph properties remains unexplored, which could be a significant aspect to consider for comprehensive temporal graph analysis.

2. The omission of key recent models like PINT[1] and ROLAND[2] from the baseline comparisons limits the evaluation's depth, potentially skewing the perception of the proposed model's performance.

3. The introduction of the Newtonian phase space model is an interesting conceptual proposition but remains unexploited in the actual modeling and theoretical justification, which can be seen as a disconnect between the proposed concepts and their practical implementation.


[1] Souza, Amauri, et al. "Provably expressive temporal graph networks." Advances in Neural Information Processing Systems 35 (2022): 32257-32269.

[2] You, Jiaxuan, Tianyu Du, and Jure Leskovec. "ROLAND: graph learning framework for dynamic graphs." Proceedings of the 28th ACM SIGKDD Conference on Knowledge Discovery and Data Mining. 2022.

**Questions:**

1) How does the algorithm perform on other prediction task in addition to the growth rate prediction. Possible metrics include temporal global efficiency, temporal-correlation coefficient, temporal betweenness centrality  and more. [1]
2) What role does the Newtonian dynamics play in the proposed algorithm?


[1] Nicosia, Vincenzo, et al. "Graph metrics for temporal networks." Temporal networks (2013): 15-40.

---

> ### Author Response · Authors · 2023-11-17
> **Author response to reviewer ehy4**
>
> We thank the reviewer for their time and valuable feedback.
>
> **W1 Focusing solely on the growth rate as a graph property is a limitation**
>
> **A:** We thank the reviewer for raising this point. Indeed exploring additional graph properties is an important direction. In this revision, we have added two additional properties: graph density and node counts growth in Appendix D, Table 10 and Table 11. We observe that GraphPulse significantly outperforms all baselines, which shows it is generalizable across multiple graph properties.
>
> **W2 The omission of key recent models like PINT[1] and ROLAND[2] from the baseline comparisons**
>
> **A:** We thank the reviewer for pointing out these references. The PINT model is designed for continuous time dynamic graphs (CTDGs), a different setting than our discrete time dynamic graph (DTDG) setting and it is non-trivial to extract graph level embedding for CTDG methods as the node embeddings are updated based on each edges (rather than each graph snapshot).
> ROLAND is a notable work in bringing static GNNs to a dynamic setting (as opposed to a specific new model). While ROLAND is a great framework for dynamic graphs, it is not trivial to compare its problem setting with our considered setting as explained below.
>
> In the live-update setting of the ROLAND, the prediction labels at timestamp $t-1$ (i.e., $y_{t-1}$) are first split into $y_{t-1}^{train}$ and $y_{t-1}^{val}$, where the $y_{t-1}^{train}$ is used for fine-tuning the model and $y_{t-1}^{val}$ is used for validation evaluation. Then, at time $t+1$ where the $y_t$ are observed, the performance of the model is tested. This process is repeated for each time interval, and the average performance over all graph snapshots is reported.
>
> It should be noted that the aforementioned evaluation procedure of ROLAND is different from our considered setting which is in line with the standard fixed split evaluation (adopted in previous works such as EvolveGCN and HTGN). In our evaluation, we construct the training, validation, and test sets by chronologically splitting the available graph snapshot sequence. We evaluate the trained model for every snapshot of the test set, however, the model is not updated based on the test observations.
>
> **W3 The introduction of the Newtonian phase space model is an interesting conceptual proposition but remains unexploited in the actual modeling and theoretical justification**
>
> **A:** We thank the reviewer for highlighting this point. By formulating the temporal graph learning problem through the lens of Newtonian space, we establish a discrete dynamical system. This framework allows us to assess our methodology using TDA Mapper through extensive experiments as detailed in Appendix A. It is important to note that a comprehensive theoretical justification for motion in a phase space, even in its simplest form, requires formulating equations for a discrete dynamical system derived from the specific context. In our approach, we opted for a more pragmatic strategy, validating our model within a known phase space and demonstrating its utility across various settings.
>
> GraphPulse undertook the crucial step of ensuring the construction of a valid and valuable phase space. We achieve this validation by using the visually explainable tool; i.e., TDA Mapper (refer to Figure 2). Building on this, the graphs can be directly put into a phase space where their trajectories can be studied. In Figure 8 of the Appendix, we show such a trajectory.
>
> **Q1 How does the algorithm perform on other prediction task in addition to the growth rate prediction?**
>
> **A:** We thank the reviewer for the reference (added to the Related Work Section) and for proposing the additional measures that are definitely interesting to explore. As mentioned above, in this rebuttal, we showed that GraphPulse is able to generalize to graph properties including density and node count. Due to the time limit of the rebuttal phase, we are considering adding other properties later in the future.
>
> **Q2 What role does the Newtonian dynamics play in the proposed algorithm?**
>
> **A:** The core idea for our temporal framework is inspired by Newton’s laws of motion and we formulate temporal graph property prediction as predicting the trajectory of a motion in a phase space. First, a novel perspective is presented by connecting temporal graph property prediction with discrete dynamical systems, a well-established mathematical field. Second, such a connection allows us to test the hypothesis that our TDA Mapper approach aids in predicting graph trajectories. To validate this, we conducted extensive experiments in various phase spaces with different settings, as detailed in Appendix A. These experiments not only support our intuition but also explain the significance of TDA Mapper in extracting complex feature information from token networks, contributing to trajectory prediction.

---

> > ### Comment · Reviewer_ehy4 · 2023-11-20
> > **Thanks for the response**
> >
> > Thank you for providing a more comprehensive evaluation with additional prediction tasks which is very helpful to understand the usefulness of the proposed methods. I am raising my score to 6.

---

> > > ### Author Response · Authors · 2023-11-21
> > > **Author response to reviewer ehy4**
> > >
> > > Thank you for your considerate review and for dedicating the time to engage in the discussion process and raise the score for our work. Your feedback is highly appreciated and helped improve this work significantly.

---

### Author Response · Authors · 2023-11-17
**Overall response to all reviewers**

We thank the reviewers for their thoughtful and constructive feedback. We are glad to see that the reviewers stated that our paper pioneers the integration of TDA into the study of temporal graphs (reviewer ehy4), is novel (reviewer NvJb), technically sound (reviewer Qyn2) and illustrates an interesting application (reviewer yFgn). In addition, the novel temporal graph property prediction task proposed in this work is well motivated (reviewer NvJb), and accessible for broad application (reviewer ehy4). In this revision, we have incorporated reviewers’ suggestions to improve our manuscript (changes marked in blue). We hope that our responses and the modifications we made address the reviewers' concerns. We would be glad to provide additional clarification and engage in further discussion.

---

### Public Comment · ~Manuel_Dileo1 · 2024-05-06
**Questions on ROLAND and live-update setting**

Hi there, thank you for your nice work! I have a couple of questions about your training and evaluation setting.

It is not completely clear - even from the comments in the rebuttal - why ROLAND has not been considered as a baseline for your work. In fact, despite being a general framework for repurposing static models into dynamic ones, You et al. proposed concrete instances of their framework that reach SOTA results on their benchmarks, which are evaluated even in the fixed split setting you adopted in your work.

A second clarification concerns the live-update setting:  since the evolving nature of the datasets you used in the evaluation, it is not clear why you only use the fixed-split setting while neglecting the live-update setting to evaluate the model. Indeed, it has been shown that the fixed split setting does not take into account the evolving nature of the data. This should be interesting to evaluate since most of the baselines in the paper (e.g. EvolveGCN, GRUGCN) were already evaluated in the live-update setting in the ROLAND paper. Lastly, some subsequent works on discrete-time dynamic graph learning adopt the live-update setting for their training and evaluation, including baselines as well (https://link.springer.com/article/10.1007/s10994-023-06475-x, https://openreview.net/forum?id=77Tyf2SFhX).

 Looking forward to hearing your insights! All the best.

---

> ### Public Comment · ~Kiarash_Shamsi1 · 2024-08-13
>
> Thank you for your thoughtful feedback and for highlighting these important considerations.
>
> Regarding the use of ROLAND as a baseline, we did not include it because ROLAND's framework involves splitting each graph snapshot into a training set and validation set at each step. Given that our focus is on graph property prediction at the graph level, we felt this approach wasn't directly applicable to our work. Our focus was on a standard setting that is closely aligned with real-world applications, where updating models continuously is often challenging due to constraints like model size, data volume, or computational complexity. Additionally, in such settings, test data may not always be reliable for real-time model updates, leading us to adopt an evaluation setup distinct from that of ROLAND.
>
> Regarding the live-update setting, we chose to concentrate on the fixed-split approach as it better reflects scenarios where continuous model updates are not always practical. We acknowledge the significance of the live-update setting, particularly in light of the strong results it has yielded in other works, including ROLAND. Our decision was based on ensuring our model's robustness and relevance in environments where continuous updates may not be feasible.
>
> We sincerely appreciate your insightful questions and hope this explanation clarifies our approach. If you would like to discuss further, please reach out to us by email. Thank you again for engaging with our work.

---

### Meta-Review · Area_Chair_e1G9 · 2023-12-04

**Metareview:**

The paper introduces GraphPulse, a novel approach for analyzing and predicting the evolution of temporal graphs through a combination of Temporal Graph Neural Networks and Topological Data Analysis. It presents a process that takes snapshots of temporal graphs at fixed intervals, constructs Topological Data Analysis Mapper representations for these snapshots, and then uses these topological features along with snapshot graph features for sequential modeling to predict future graph properties. The paper conducts experiments that show superior performance over existing models in predicting network growth on several datasets.

Reviewers have raised concerns about the paper's presentation and the experimental setting. However, the value of the contributions outweighs these concerns. Acceptance is recommended, and integrating the reviewers' feedback in the preparation of the final version of the paper is encouraged.

**Justification For Why Not Higher Score:**

The reviewers unanimously believe the paper should be accepted. However, the paper still suffers from several drawbacks, including:

1. The proposed method only works for discrete-time dynamic graph models, whereas the continuous-time dynamic graph model is more generalized and reasonable.
2. The presentation needs improvement, as some reviewers are unable to comprehend some of the technical details in the paper.

Therefore, I recommend that the paper only be presented as a poster.

**Justification For Why Not Lower Score:**

The reviewers unanimously believe the paper should be accepted.

---

### Decision · Program_Chairs · 2024-01-16

Accept (poster)